# GraphArena: Evaluating and Exploring Large Language Models on Graph Computation

**Jianheng Tang, Qifan Zhang, Yuhan Li, Nuo Chen, Jia Li**[*]
Hong Kong University of Science and Technology (Guangzhou)
`jtangbf@connect.ust.hk, jialee@ust.hk`

## Abstract

The "arms race" of Large Language Models (LLMs) demands new benchmarks to examine their progresses. In this paper, we introduce GraphArena, a benchmarking tool designed to evaluate LLMs on real-world graph computational problems. It offers a suite of four polynomial-time tasks (e.g., Shortest Distance) and six NP-complete challenges (e.g., Traveling Salesman Problem). GraphArena features a rigorous evaluation framework that classifies LLM outputs as correct, suboptimal (feasible but not optimal), hallucinatory (properly formatted but infeasible), or missing. Evaluation of over 10 LLMs reveals that even top-performing LLMs struggle with larger, more complex graph problems and exhibit hallucination issues. We further explore four potential solutions to address this issue and improve LLMs on graph computation, including chain-of-thought prompting, instruction tuning, code writing, and scaling test-time compute, each demonstrating unique strengths and limitations. GraphArena complements the existing LLM benchmarks and is open-sourced at `https://github.com/squareRoot3/GraphArena`.

## 1 Introduction

Evaluating LLMs, especially their advanced reasoning capabilities, remains a significant long-term challenge. While standardized tests in fields like mathematics (Hendrycks et al., 2021; Cobbe et al., 2021), medicine (Chen et al., 2023; Li et al., 2023a), and multi-task collections (Hendrycks et al., 2020; Srivastava et al., 2022) have become popular LLM benchmarks, concerns about data leakage in pretraining corpora (Aiyappa et al., 2023; Xu et al., 2024) raise questions about whether LLMs are genuinely reasoning or merely memorizing information. Crowd-sourced evaluations can address this issue but are typically expensive and time-consuming (Zheng et al., 2023b; Chiang et al., 2024; Chang et al., 2024). Alternatively, evaluations using synthetic data, such as theorem proving (Trinh et al., 2024; Xin et al., 2024), offer more rigorous assessments but often lack real-world relevance.

To balance factors such as cost, novelty, complexity, diversity, and practical applicability in LLM evaluation, the research community has expanded its focus beyond text to include various modalities, such as vision (Lu et al., 2024), tabular (Sui et al., 2024), and spatiotemporal data (Lopez-Lira & Tang, 2023). Among these modalities, *graphs* emerge as a particularly promising avenue for evaluating LLMs' capabilities in several key areas: interpreting relational information or structured knowledge, processing non-sequential or non-Euclidean data, and generalizing across diverse structures or tasks. The potential of graph-based evaluations for LLMs has been recognized and explored in recent literature (Perozzi et al., 2024; Wang et al., 2023; Chen et al., 2024b).

In this paper, we benchmark LLMs on *graph computational problems*, which often require systematic traversal or search algorithms to solve. These problems serve as an ideal testbed for evaluating the reasoning capabilities of LLMs (Li et al., 2024; Wei et al., 2024; Chen et al., 2024a). While several graph problem-solving benchmarks have been established, such as NLGraph (Wang et al., 2023) and GraphQA (Fatemi et al., 2023), as well as algorithmic reasoning datasets like CLRS-Text (Markeeva et al., 2024) and MAGMA (Taylor et al., 2024), we identify several limitations in the existing landscape. First, these datasets are predominantly built on synthetic graphs, such as those generated by the Erdős-Rényi (ERDdS & R&wi, 1959) models, which may not adequately reflect real-world diversity. Second, the tasks within these benchmarks are generally confined to basic structural

---

[*]Corresponding author.

understanding (e.g., connectivity checks) and algorithm execution (e.g., breadth-first search). They largely overlook the assessment of higher-order reasoning skills, such as problem abstraction and simplification, strategy comparision and selection, etc. These capabilities are better demonstrated through solving more complex and open-ended tasks, such as NP-complete problems. Third, current evaluation methods typically rely on string matching between the final answers and the responses, allowing models to succeed through guesswork rather than demonstrating a genuine understanding of the underlying logic. Such methods lack a nuanced categorization of failure modes, limiting the depth of insights derived from the metrics.

To address these issues, we introduce **GraphArena** in Section 2, a benchmarking tool designed to assess the reasoning capabilities of LLMs when tackling graph computational problems. Compared with previous graph benchmarks, three core improvements are introduced:

- **Realistic Graph Collection.** Each problem in GraphArena features a subgraph sampled from a rich assortment of real-world graphs, including knowledge graphs, social networks, and molecular structures. Our sampling process is designed to preserve the original graph topology and attributes to capture real-world diversity. Additionally, each problem is contextualized within the real-world setting of the graph, offering a more authentic and challenging evaluation environment.

- **Comprehensive Task Selection.** Our benchmark presents more demanding graph tasks to assess multi-dimensional reasoning skills. As shown in Figure 1, it encompasses four polynomial-time challenges that test *direct* algorithmic reasoning, requiring models to accurately execute algorithm procedures step-by-step. Moreover, it includes six NP-complete task that demand *meta* algorithmic planning, where models must analyze problem characteristics and select suitable algorithms to excute. This selection of diverse and well-known tasks ensures a thorough evaluation of LLMs' abilities to perform both straightforward algorithm execution and strategic decision-making.

- **Rigorous Path-based Evaluation.** We propose a detailed evaluation protocol for each task to differentiate genuine problem-solving from pattern-based guessing. LLMs must generate the entire solution path (or its components) within the graph that leads to the solution. Through a three-step process involving path extraction, feasibility check, and optimality verification, responses are categorized as correct, suboptimal (feasible but not optimal), hallucinatory (properly formatted yet infeasible), or missing. This approach allows for a nuanced and accurate comparison among LLMs.

In our experiments, we extend the comparative analysis beyond LLMs by incorporating a diverse set of baseline methods, including classical graph algorithms, graph neural networks, and Graph-LLM hybrid approaches. Additionally, we explore strategies beyond conventional prompt engineering to improve LLMs' performance in graph reasoning tasks. Our study seeks to address two key research questions:

**To what degree can current LLMs solve graph computational problems? (Section 3.1)** After evaluating ten popular LLMs on 10,000 GraphArena problems, we find that LLMs perform significantly better on polynomial-time tasks than on NP-complete ones. They excel in direct algorithmic reasoning but struggle with meta algorithmic planning, which demands higher-level strategic thinking. Even the most advanced LLMs, such as GPT-4o and Claude-3.5-Sonnet, struggle with graph computations—often producing hallucinatory outputs—particularly when handling larger graphs, more complex tasks, particularly with larger graphs and more complex tasks. For NP problems, these models typically default to greedy algorithms, though they sometimes utilize more effective approximation techniques. Despite their current limitations, our results highlight the potential of LLMs to serve as alternative heuristics for NP tasks, complementing existing approximation methods.

**How can we further enhance LLMs on graph reasoning? (Section 3.2)** While previous benchmarks for graph problems have primarily focused on prompt engineering, we observe that chain-of-thought prompting exhibits limited effectiveness on GraphArena. To overcome this limitation, we explore several advanced approaches: (1) **Instruction tuning** on similar graph problems and solution paths enhances performance on small-scale polynomial tasks. However, it proves less effective for large-scale NP problems, as fine-tuning alone is insufficient to develop complex reasoning abilities, such as algorithmic planning and long-range reasoning. (2) **Code generation and execution** by LLMs show significant promise for solving large graphs and tackling more complex tasks, though some performance degradation is observed on small graphs. (3) **Increasing test-time compute** and allowing LLMs multiple attempts yield modest but consistent performance improvements. These findings underscore the need for innovative strategies to advance LLMs' graph reasoning capabilities.

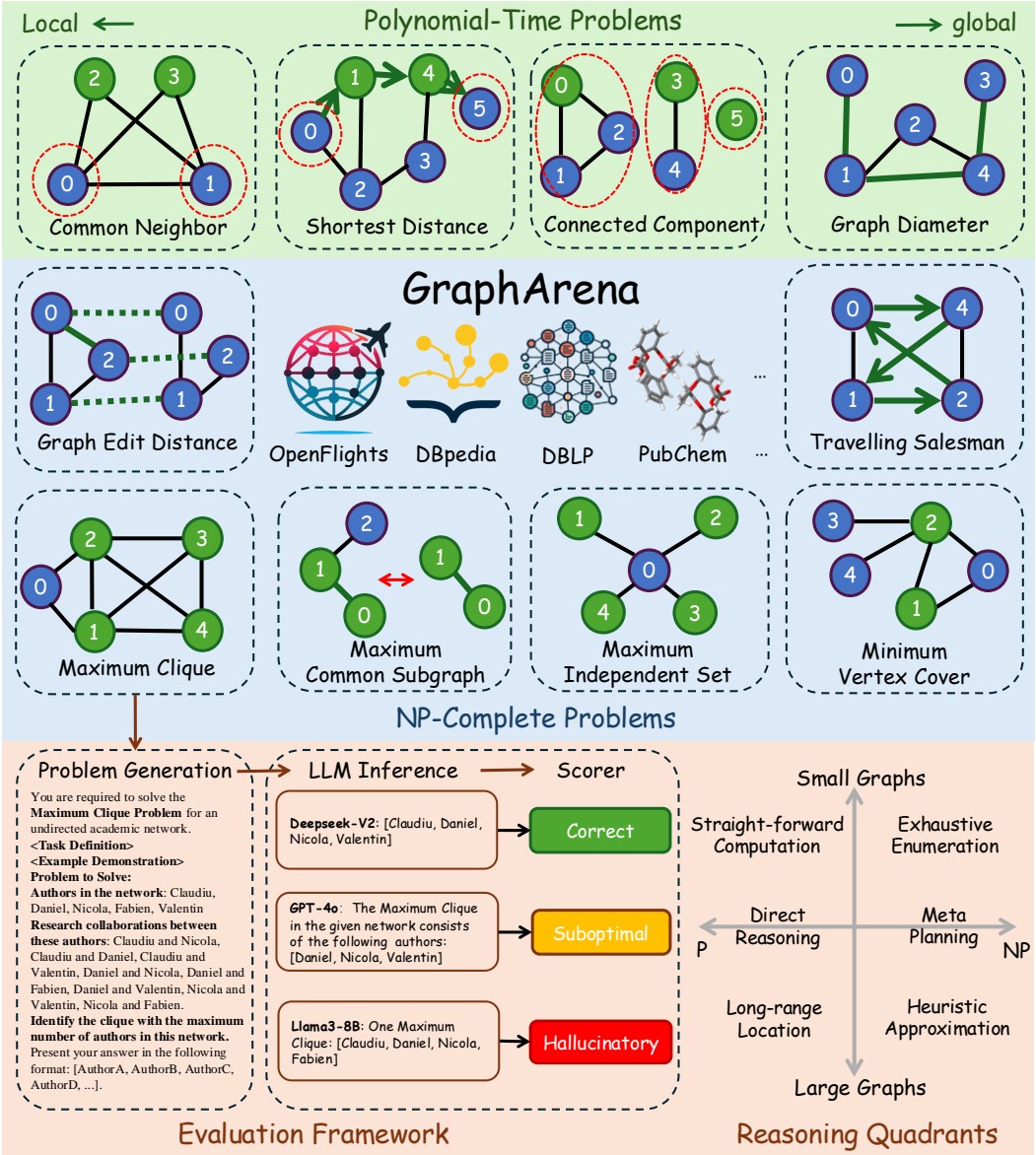

Figure 1: Overview of the GraphArena benchmark.

## 2 BENCHMARK CONSTRUCTION

Figure 1 provides an overview of our proposed GraphArena benchmark. In this section, we first discuss the collection and sampling methodologies for real-world graphs in Section 2.1, then offers detailed descriptions of 10 tasks within GraphArena and the problem generation process in Section 2.2. Finally, Section 2.3 describes the evaluation framework to score and compare LLMs' responses.

### 2.1 DATASET COLLECTION

GraphArena distinguishes itself from previous benchmarks by utilizing real-world graphs, offering richer diversity compared with synthetic ones. Graphs are collected from five sources: DBLP (Ley, 2002), an academic collaboration network with over 1.3 million nodes (authors) and 5 million edges (collaborations); Social Network (Rossi & Ahmed, 2015), a composite graph combining Digg, Flixster, Lastfm, and Pokec data, totaling over 6 million nodes (users) and 40 million edges (friendships); DBpedia (Bizer et al., 2009), a knowledge graph subset (DBpedia1M (Xin et al., 2022)) with over 1 million nodes (entities) and 7 million edges (relationships); OpenFlights (OpenFlights),

an airport network consisting of 3,390 nodes (airports) and 19,166 edges (flight routes) weighted by geographical distance; and PubChemQC (Nakata & Shimazaki, 2017), containing 3.7 million molecular graphs from the PCQM4Mv2 Dataset (Hu et al., 2021), where nodes represent atoms and edges represent chemical bonds. The detailed description of these datasets is in Appendix A.

We employ a random walk with restart strategy initiating from randomly selected nodes to sample subgraphs effectively. This enables us to identify the top-$k$ most frequently visited nodes, forming local dense subgraphs that closely maintain the topological features of the original graphs. For the PubChemQC dataset, we sample molecular graphs directly by specified sizes. All graphs are treated as unweighted and undirected, except for OpenFlights which is weighted.

## 2.2 Task Selection

While there is no gold standard for task selection, we focused on well-known tasks with rich real-world applications, and challenging tasks that assess multi-dimensional reasoning skills. We consider four polynomial-time tasks to test *direct algorithmic reasoning*, requiring models to accurately execute established algorithm procedures step-by-step:

- **Common Neighbor:** For a graph $\mathcal{G} = \{\mathcal{V}, \mathcal{E}\}$ with vertices $v_1, v_2 \in \mathcal{V}$, the task involves identifying the subset $S \subseteq \mathcal{V}$ where each $u \in S$ is connected to both $v_1$ and $v_2$. The objectives are to (1) identify these common neighbors, and (2) maximize their count. We utilize the DBLP academic collaboration graph for this task.
- **Shortest Distance:** In a graph $\mathcal{G} = \{\mathcal{V}, \mathcal{E}\}$, determine the shortest path from node $v_1$ to $v_2$. The goals are to (1) find a viable path $(v_1, u_1, \ldots, v_2)$, and (2) minimize its hop count. We apply this task to the DBpedia knowledge graph.
- **Connected Component:** In this task, the model identifies one representative node from each connected component within a graph. For a given graph $\mathcal{G} = \{\mathcal{V}, \mathcal{E}\}$, the objectives are to (1) identify a set of vertices such that each is from a distinct connected component, and (2) ensure that these vertices represent all possible connected components of the graph. This task is applied to the Social Network dataset.
- **Graph Diameter:** The diameter of a graph is the maximum distance between any pair of nodes in the graph. Given a graph $\mathcal{G} = \{\mathcal{V}, \mathcal{E}\}$, this task entails finding (1) a shortest path between two arbitrary vertices and (2) ensuring it is the maximum possible among all shortest paths. This is performed using the DBpedia knowledge graph.

Each task is briefly described by a keyword (Neighbor, Distance, Component, Diameter) to facilitate easy reference. We decompose each task into two requirements, the first basic requirement and the second optimal requirement. We further consider six NP-complete tasks to evaluate *meta algorithmic planning*, where models must analyze problem characteristics and select suitable algorithms to excute:

- **Maximum Clique Problem (MCP):** Identify the largest complete subgraph, or clique, in a given graph $\mathcal{G} = \{\mathcal{V}, \mathcal{E}\}$ sampled from DBLP. The task entails (1) finding a clique $\mathcal{C} \subseteq \mathcal{V}$ that is feasible within the graph, and (2) ensuring that $\mathcal{C}$ is the largest among all possible cliques.
- The descriptions for the remaining five tasks—**Maximum Independent Set (MIS)**, **Minimum Vertex Cover (MVC)**, **Maximum Common Subgraph (MCS)**, **Graph Edit Distance (GED)**, and **Traveling Salesman Problem (TSP)**—are provided in Appendix B due to space constraints.

**Problem Generation:** For each of the 10 tasks, we randomly sample 500 small and 500 large graphs to create two distinct subsets, yielding a total of 10,000 graphs. For difficulty calibration, we use task-specific graph scales determined by each problem's computational complexity. For the Neighbor and Distance tasks, small graphs contain 4 to 19 nodes, while large graphs contain 20 to 50 nodes. For Component, Diameter, MCP, MIS, and MVC, small graphs have 4 to 14 nodes, and large graphs have 15 to 30 nodes. For MCS, GED and TSP, small graphs have 4 to 9 nodes and large graphs have 10 to 20 nodes. The ground truth of each problem is generated by corresponding exact algorithms.

After sampling graphs of the given size, we encode graph problems into text using templates, a common practice in previous benchmarks (Wang et al., 2023; Chen et al., 2024a). An example of this encoding is shown in the bottom left of Figure 1. The process begins with an introduction and definition of the task, followed by a randomly selected example with the correct answer to demonstrate the problem-solving process and task requirements. Subsequently, the problem to be solved is presented with detailed graph information. While the graphs in each problem may not

appear large, their text representation often results in problems containing up to 6,000 tokens, posing a long-context challenge. Detailed statistics on graph sizes and problem lengths, along with examples for each task, are provided in Table 4 and Appendix B.

## 2.3 EVALUATION PROCESS

We develop a fine-grained evaluation process to assess the reasoning process of LLMs and prevent pattern-based guesses. For each task, LLMs are required to output the relevant graph component or path that contributes to the final answer, as highlighted by the green elements in each graph shown in Figure 1. Our evaluation process consists of three steps: (1) Path Extraction: We use regular expressions to extract the proposed solution path from the LLM's response. If a valid path cannot be extracted due to formatting issues, lack of response, etc., the response is categorized as *missing*. (2) Feasibility Check: We employ scripts to verify whether the extracted path meets the basic requirements of the problem. Paths that fail this check are categorized as *hallucinatory*. (3) Optimality Verification: For feasible paths, we calculate a path score (e.g., total route length for TSP, and number of cliques for MCP). This score is then compared to the optimal solution obtained through exhaustive search. If they match, the result is *optimal*; otherwise, it is considered *suboptimal*.

For example, as shown in the bottom middle of Figure 1, Deepseek-V2 correctly identifies the maximum clique of four authors. In contrast, GPT-4o finds a feasible but suboptimal clique with three nodes. Llama3-8b produces a hallucinatory response, as the nodes it chooses do not form a clique. If we were to evaluate only numerical results, as in existing benchmarks, this error would be overlooked since it also leads to the correct number "4".

**Metrics.** Given the rarity of missing results in our experiments (less than 1% in most cases), our primary metrics are **Accuracy** (the proportion of correct answers), **Feasibility** (the proportion of both correct and sub-optimal answers), and **Hallucination** (the proportion of hallucinatory responses). We also introduce ranking-based metrics including for more comprehensive comparison between different LLMs, particularly useful when most responses are suboptimal. These include Mean Reciprocal Rank (**MRR**), **Top-1** Probability, and **Top-3** Probability.

## 3 EXPERIMENTS

**Experimental Setup.** Our evaluation in GraphArena encompassed ten prominent LLMs, including both closed-source and open-source variants, as well as single models and mix-of-experts architectures of varying scales. Closed-source models included the GPT-4o-2024-08-06 (OpenAI, 2023), GPT-4o-mini-2024-07-18, Claude-3.5-sonnet-20241022 (Anthropic, 2024), and GLM-4-plus (GLM, 2024) (whose smaller version is open-sourced). Open-source models include Llama3-70b-instruct and Llama3-8b-instruct from Meta (AI@Meta, 2024), Qwen2.5-72b-instruct from AliCloud (Bai et al., 2023), and Gemma-7b-it from Google Cloud (7b it). We also evaluated two mix-of-expert LLMs: Deepseek-V2 (Bi et al., 2024) with 230 billion parameters (21 billion active) and Mixtral-7x8b (Jiang et al., 2024) with 47 billion parameters (13 billion active). We utilized a low temperature setting of 0.1, imposed no constraints on output length, and maintained other configurations default for all LLM models. The full-parameter fine-tuning process was conducted with a learning rate of 0.0001, using a cosine learning rate scheduler with a warmup ratio of 0.1, and a batch size of 4.

For closed-source LLMs and open-source models exceeding 10 billion parameters, we utilized the corresponding cloud-based services. Smaller open-source models were run on our local infrastructure equipped with four NVIDIA H800 PCIe 80GB GPUs. Given the computational demands and costs associated with LLM evaluation, we conducted a single run per model across the 10,000 problems in GraphArena. Additionally, we compared LLM performance against *traditional graph algorithms*, *graph neural networks*, and *Graph-LLM hybrid models* (Chen et al., 2024a; Perozzi et al., 2024) to provide a comprehensive assessment of LLMs' capabilities in graph computation. For these models, we follow the original implementations and configurations. All problems, responses, and our codebase have been recorded and will be open-sourced to facilitate reproducibility.

Table 1: Average rankings and performance of 10 LLMs on 4 Polynomial-time and 6 NP-complete tasks across small and large graphs. All metrics excluding MRR are scaled to [0, 100]. **Acc.**, **Fea.**, and **Hallu.** represent Accuracy, Feasibility, and Hallucination probability, respectively. For all metrics except Hallucination, higher values indicate better performance. The best-performing open and closed-source models are highlighted in **bold**.

| LLM | Polynomial-Time Tasks (Small) | | | | | | Polynomial-Time Tasks (Large) | | | | | |
| --- | --- | --- | --- | --- | --- | --- | --- | --- | --- | --- | --- | --- |
| | MRR | Top-1 | Top-3 | Acc. | Fea. | Hallu. | MRR | Top-1 | Top-3 | Acc. | Fea. | Hallu. |
| Claude-3.5-sonnet | **0.87** | **84.3** | **86.2** | **82.2** | **86.5** | 13.4 | **0.72** | **67.2** | **71.7** | **58.7** | **70.5** | 29.5 |
| GPT-4o | 0.83 | 79.5 | 82.3 | 76.9 | 84.6 | 13.4 | 0.58 | 49.7 | 56.6 | 43.5 | 58.9 | 37.3 |
| GPT-4o-mini | 0.76 | 70.6 | 75.1 | 68.9 | 82.8 | 16.1 | 0.50 | 40.7 | 48.3 | 36.6 | 56.9 | 40.7 |
| GLM-4-plus | 0.79 | 75.1 | 77.3 | 72.7 | 78.8 | 18.6 | 0.60 | 53.1 | 58.6 | 45.7 | 57.0 | 42.0 |
| Llama3-70b | **0.68** | **63.2** | **66.3** | **61.2** | 71.9 | 27.3 | 0.47 | 37.0 | 44.3 | 31.6 | 49.3 | 49.0 |
| Llama3-8b | 0.37 | 29.2 | 30.8 | 28.5 | 44.5 | 53.9 | 0.21 | 10.8 | 12.6 | 9.4 | 18.9 | 78.2 |
| Qwen2.5-72b | 0.67 | 59.9 | 64.0 | 59.0 | **74.8** | 11.7 | **0.51** | **42.1** | **49.4** | **39.9** | **55.7** | 36.3 |
| Deepseek-v2 | 0.60 | 53.1 | 56.9 | 51.4 | 64.2 | 35.0 | 0.39 | 29.0 | 35.7 | 24.7 | 37.9 | 61.1 |
| Mixtral-7x8b | 0.46 | 38.5 | 40.4 | 37.8 | 51.3 | 46.4 | 0.27 | 15.8 | 19.9 | 13.8 | 31.5 | 67.2 |
| Gemma-7b | 0.34 | 25.5 | 26.7 | 25.2 | 38.6 | 61.4 | 0.19 | 9.6 | 10.1 | 9.2 | 16.1 | 84.0 |
| | NP-Complete Tasks (Small) | | | | | | NP-Complete Tasks (Large) | | | | | |
| Claude-3.5-sonnet | **0.65** | **57.5** | **65.7** | **47.8** | 74.4 | 25.5 | **0.46** | **32.7** | 48.4 | **7.2** | 51.9 | 47.0 |
| GPT-4o | **0.65** | 55.6 | 66.5 | 47.3 | 77.0 | 20.2 | 0.41 | 24.2 | 46.4 | 6.3 | **53.4** | 43.6 |
| GPT-4o-mini | 0.56 | 46.5 | 56.4 | 39.2 | 72.4 | 25.5 | 0.27 | 12.8 | 24.9 | 3.3 | 39.7 | 55.4 |
| GLM-4-plus | 0.60 | 49.3 | 60.5 | 41.3 | **78.8** | 20.4 | 0.35 | 18.7 | 39.4 | 4.8 | 50.5 | 49.1 |
| Llama3-70b | **0.54** | **43.8** | **52.6** | **36.8** | 75.3 | 24.5 | **0.30** | **15.1** | 27.8 | **4.7** | **45.0** | 55.0 |
| Llama3-8b | 0.35 | 23.6 | 29.3 | 20.2 | 57.9 | 41.7 | 0.20 | 8.4 | 14.6 | 1.9 | 24.1 | 75.5 |
| Qwen2.5-72b | 0.39 | 26.7 | 34.7 | 20.6 | 65.1 | 27.0 | 0.20 | 7.6 | 15.2 | 0.7 | 21.4 | 42.1 |
| Deepseek-v2 | 0.52 | 40.1 | 51.1 | 33.7 | 74.1 | 24.5 | 0.29 | 14.8 | 27.7 | 3.1 | 40.9 | 58.1 |
| Mixtral-7x8b | 0.30 | 18.1 | 23.9 | 15.6 | 59.3 | 37.4 | 0.20 | 6.1 | 14.4 | 1.6 | 29.4 | 64.6 |
| Gemma-7b | 0.28 | 15.6 | 23.2 | 12.9 | 57.1 | 42.6 | 0.22 | 7.5 | 20.4 | 0.9 | 31.2 | 68.8 |

## 3.1 MAIN RESULTS

In Table 1, we present the average performance of ten LLMs across four polynomial-time and six NP-complete tasks on both small and large graphs. Claude-3.5-sonnet consistently ranks as the top performer across most metrics and settings, while Llama3-70b is the leading open-source model. Performance generally declines for all LLMs as graph sizes increase and tasks transition from P to NP, significantly expanding the solution space. Notably, all LLMs struggle considerably with NP-complete tasks on large graphs, achieving less than 10% accuracy. This indicates that the most challenging scenarios in GraphArena still exceed the current capabilities of LLMs.

GraphArena also reveals a significant performance disparity between LLMs of varying parameter sizes. For instance, in polynomial-time tasks, the gap between Llama3-70b and Llama3-8b is substantial—61.2% vs. 28.6% on small graphs and 31.6% vs. 9.4% on large graphs. This gap is significantly wider than in existing benchmarks, such as GSM8K (93.0% vs. 79.6%) and GPQA (39.5% vs. 34.2%). A similar trend is observed in the Qwen series. Interestingly, the performance gap between GPT-4o and GPT-4o-mini is not as large as one might expect given their difference in API pricing. Moreover, models with fewer parameters are more prone to hallucination, with ratios of 78.2% for Llama3-8b and 84.0% for Gemma-7b, indicating that graph computation necessitate more advanced reasoning capabilities that larger LLMs are better equipped to manage.

To closely examine model performance on individual tasks, we compare the feasibility and accuracy of five selected LLMs on each task in Figure 2. Results for the remaining five LLMs are presented in Figure 8 in Appendix C. Model performance varies substantially across tasks, with tasks like Diameter, MCS, and MVC demonstrating notably low feasibility, indicating high hallucination rates.

**Hallucination Impact.** We illustrate the hallucination probability against varying node sizes for three selected tasks in Figure 4, with results for the remaining seven tasks presented in Figures 9 and 10 in Appendix C. These plots reveal a significant, nearly monotonic increase in hallucination ratio as node size grows from 5 to 30. For example, in the Diameter task, GPT-4o's hallucination probability rises dramatically from 16% at a node size of 5 to more than 80% at a node size of 30. This trend underscores that larger graph sizes, and their associated challenges in long-range multi-step reasoning, are major contributors to hallucination.

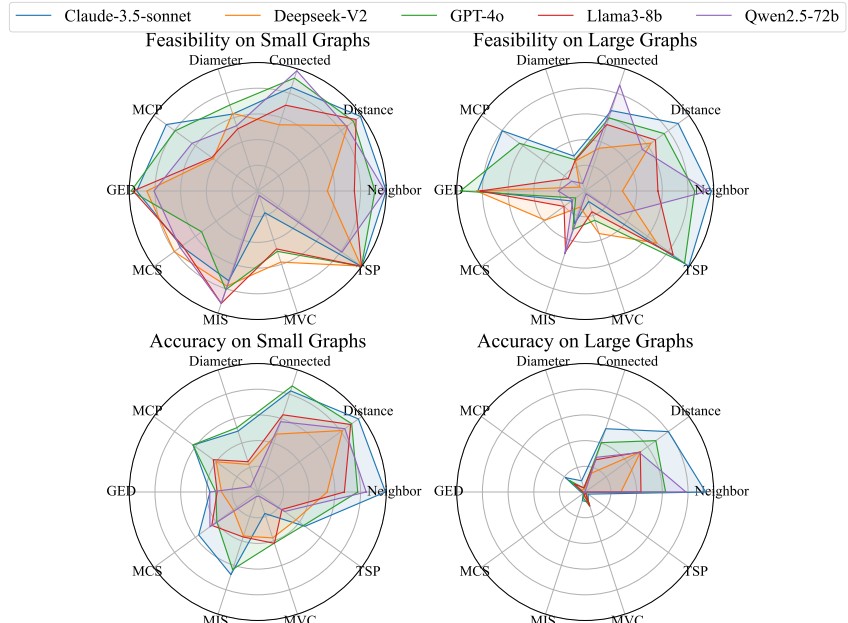

Figure 2: Feasibility and accuracy comparison of five selected LLMs on each individual task. The circles represent performance levels, progressing outward from 20% to 100% in increments of 20%.

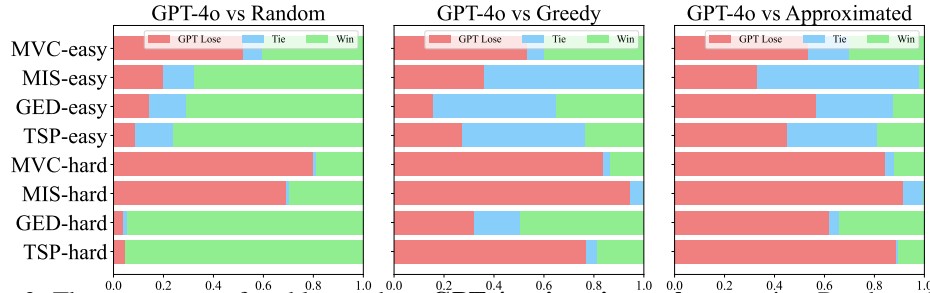

Figure 3: The percentage of problems where **GPT-4o wins, ties, or loses** against Random, Greedy, and Approximated algorithms.

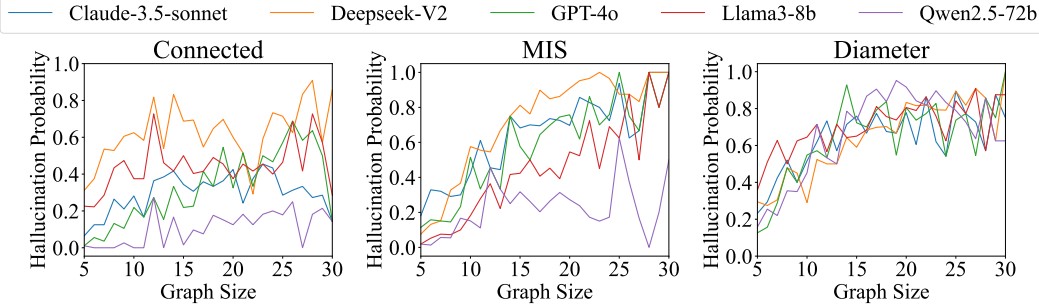

Figure 4: The influence of graph size on hallucination probability for the Maximum Independent Set, Graph Diameter, and Connected Component tasks.

**Comparison with Graph Algorithms.** In GraphArena, we compute the ground truth for each problem using exact graph algorithms and evaluate LLMs' accuracy against these optimal solutions. However, as algorithms can take hours to find optimal solutions for some NP-complete tasks, we also compare LLMs with three suboptimal yet more efficient graph algorithms: (1) Random algorithms: randomly generate a feasible solution; (2) Greedy algorithms: choose the best option at each step; and (3) Approximation algorithms: advanced graph algorithms with problem-specific heuristics or approximations that typically outperform greedy algorithms (Christofides, 2022; Zeng et al., 2009; Boppana & Halldórsson, 1992; Bar-Yehuda & Even, 1985).

Figure 3 illustrates the comparative performance of GPT-4o against Random, Greedy, and Approximation algorithms on four selected NP-complete tasks, showing the percentage of problems where

Table 2: Performance comparison of LLMs finetuned on graph problems (-SFT).

| Task & Graph Scale | Polynomial (Small) | | | Polynomial (Large) | | | NP (Small) | | | NP (Large) | | |
|---|---|---|---|---|---|---|---|---|---|---|---|---|
| LLM | Acc. | Fea. | Hallu. | Acc. | Fea. | Hallu. | Acc. | Fea. | Hallu. | Acc. | Fea. | Hallu. |
| Llama3-8b | 28.6 | 45.3 | 53.9 | 20.2 | 58.1 | 41.7 | 9.4 | 19.7 | 78.2 | 1.4 | **31.2** | 68.8 |
| **Llama3-8b-SFT** | **54.7** | **72.5** | **17.3** | **38.1** | **65.0** | **31.1** | **38.1** | **65.0** | **31.1** | **7.5** | 25.5 | **49.6** |
| Qwen2-7b | 20.4 | 33.6 | 62.9 | 9.7 | 17.8 | 73.4 | 21.5 | **65.0** | 33.6 | 1.5 | **29.8** | 68.2 |
| **Qwen2-7b-SFT** | **84.3** | **90.6** | **9.4** | **42.0** | **59.7** | **39.5** | **39.5** | 57.6 | **15.0** | **6.2** | 26.8 | **50.5** |

Table 3: Performance comparison of LLMs prompted to write and execute code (-Coder).

| Task & Graph Scale | Polynomial (Small) | | | Polynomial (Large) | | | NP (Small) | | | NP (Large) | | |
|---|---|---|---|---|---|---|---|---|---|---|---|---|
| LLM | Acc. | Fea. | Hallu. | Acc. | Fea. | Hallu. | Acc. | Fea. | Hallu. | Acc. | Fea. | Hallu. |
| GPT-4o | **76.9** | **84.6** | **13.4** | 43.5 | 58.9 | 37.3 | **47.3** | 77.0 | 20.2 | 6.3 | **53.4** | 43.6 |
| **GPT-4o-Coder** | 62.7 | 71.7 | 18.9 | **51.1** | **62.0** | **23.0** | 46.5 | **77.4** | **16.8** | **7.9** | 46.8 | **35.4** |
| DeepSeek-V2 | 51.4 | 64.2 | 35.0 | 24.7 | 37.9 | 61.1 | **33.7** | **74.1** | 24.5 | 3.1 | **40.9** | 58.1 |
| **DeepSeek-V2-Coder** | **60.9** | **69.7** | **16.7** | **38.0** | **44.6** | **31.4** | 31.2 | 63.1 | **24.4** | **3.5** | 23.4 | **37.5** |

GPT-4o wins, ties, or loses. Comprehensive results for all LLMs are summarized in Table 5 in Appendix C. Our analysis reveals that GPT-4o consistently outperforms random solutions across most scenarios. When compared to greedy algorithms, GPT-4o demonstrates comparable performance on small-scale graphs but struggles with larger, more complex structures. Although GPT-4o rarely surpasses advanced approximation algorithms, its occasional successes hint at the potential of LLMs as alternative heuristics for NP-complete tasks.

**Comparison with Graph Neural Networks (GNNs).** We conducted a preliminary comparison with three types of graph neural networks (GNNs)—GIN (Xu et al., 2019), GAT (Veličković et al., 2017), and GraphSAGE (Hamilton et al., 2017)—as detailed in Table 6 of Appendix C. Our results indicate that Claude-3.5-sonnet generally outperforms these GNNs when they lack task-specific architecture design. Beyond this comparison, LLMs offer unique advantages beyond raw performance metrics. Their ability to interpret and respond to natural language problem descriptions significantly enhances accessibility for non-expert users, and their adaptability allows them to tackle new problem variations without explicit reprogramming.

**Comparison with Graph-LLM hybrid models.** We evaluated GraphArena against two types of graph-enhanced LLMs: GraphWiz (Chen et al., 2024a), a Llama2-7b-based model finetuned for graph reasoning, and GraphToken (Perozzi et al., 2024), a hybrid model that uses Graph Isomorphism Network (GIN) as a graph tokenizer for LLM. GraphWiz performed poorly on GraphArena (0.2% to 1.2% accuracy on small-scale tasks), primarily due to the lack of task

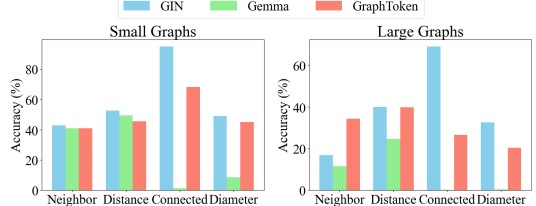

Figure 5: Performance comparison of GraphToken and its backbone LLM and GNN.

overlap and differences in graph formats between GraphArena and the GraphWiz training corpus. GraphToken generally performed between standalone GIN and Gemma-7b when fine-tuned on a comparable training set. As shown in Figure 5, it improved upon the original LLM backbone but underperformed compared to the GIN encoder, with mutual benefits only observed in the large-scale Neighbor task. These results suggest that significant improvements are needed to fully leverage the strengths of both graph-based models and LLMs.

## 3.2 EXPLORING STRATEGIES TO ENHANCE LLMS ON GRAPH COMPUTATION

Several strategies have been proposed to mitigate hallucinations in LLMs and boost reasoning capabilities (Bang et al., 2023; Ji et al., 2023; Zhang et al., 2023; Wang et al., 2022), but their effectiveness in graph computation remains to be explored. In this section, we consider the following four directions to reduce hallucination and improve LLMs on graph problem-solving:

**Chain-of-Thought (CoT) Prompting.** CoT prompting encourages models to generate intermediate reasoning steps towards a conclusion (Wei et al., 2022; Feng et al., 2024; Merrill & Sabharwal, 2023). We designed manually crafted examples for the Graph Diameter and Connected Component tasks, evaluating the impact of zero to four CoT examples. Figure 6 demonstrates a general improvement in

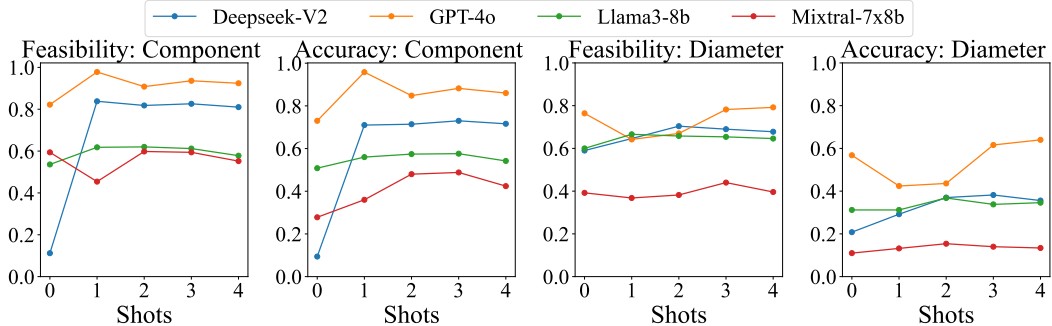

Figure 6: The influence of chain-of-thoughts with k-shot step-by-step demonstrations on the feasibility and accuracy of model performance for the Graph Diameter and Connected Component tasks.

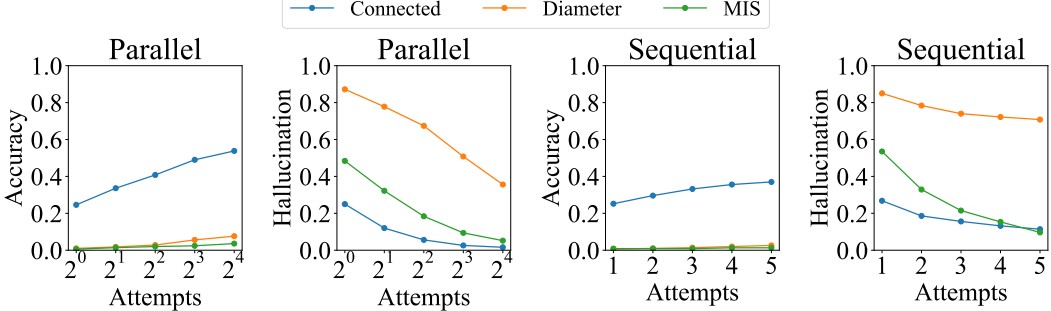

Figure 7: Performance comparison of GPT-4o-mini on three large-scale graph tasks using parallel and sequential test-time compute scaling techniques. The x-axis represents the number of attempts.

model accuracy. Deepseek-V2's performance significantly improves after adding one CoT example for the Component task. However, GPT-4o and Mixtral-7x8b show marginal improvements, and in some cases, performance degrades with more examples—a phenomenon also observed in previous benchmarks (Wang et al., 2023; Fatemi et al., 2023). These results suggest that while CoT prompting helps reduce errors, it remains insufficient to resolve hallucinations.

**Code Writing.** LLMs often struggle with mathematical reasoning, which is crucial for solving complex graph problems. To address this, we explored using external tools for computation. Inspired by the approach of writing code for math problem-solving (Gou et al., 2024), we prompted LLMs, specifically GPT-4o and Deepseek-Coder (Zhu et al., 2024), to write and execute code for solving problems in GraphArena. Results in Table 2 show that this code-writing approach effectively reduced hallucination rates, particularly on large graphs. For instance, Deepseek-V2-Coder's hallucination rate for polynomial complexity tasks on large graphs decreased from 61.1% to 31.4%. However, we observed some negative impacts on small graphs, likely due to errors in extracting graph information from text and challenges in code writing.

**Instruction Tuning.** LLMs often hallucinate on long-tail knowledge with limited training data (Mallen et al., 2023; Kandpal et al., 2023), and graph problem-solving likely falls into this category. To provide more extensive data exposure, we fine-tuned Llama3-8b and Qwen2-7b on an additional 10,000 GraphArena problems, using ground-truth solution paths as supervision. As shown in Table 2, Llama3-8b-SFT demonstrates substantial improvements over its base model, with performance even comparable to Llama3-70b. However, the improvement are more significant for small graphs and polynomial-time tasks, while long-context multi-step reasoning remains a challenge.

**Scaling Test-time Compute.** Recent studies demonstrate that scaling LLMs' test-time compute optimally can be more effective than scaling model parameters (OpenAI, 2024; Snell et al., 2024). Graph computation problems with high complexity are particularly suitable for this approach, as they often require systematic search and are challenging to solve in a single attempt. We explore two methods for scaling test-time compute: (1) *repeated sampling* (Brown et al., 2024), referred to as *parallel*, which generates multiple LLM answers and uses a checker to identify the best one; and (2) *progressive-hint prompting* (Zheng et al., 2023a), labeled as *sequential*, which enables iterative interactions between checkers and LLMs, using previous answers and results as hints to guide the

model toward correct solutions. We apply these techniques to GPT-4o-mini on three large-scale graph tasks, with results shown in Figure 7. While the hallucination ratio decreases significantly with increasing attempts, accuracy improvements are limited to simpler tasks like Connected Component. This suggests that scaling test-time compute alone is insufficient for complex graph tasks due to the expansive search space involved.

## 4 RELATED WORK

**LLM Evaluation.** Early evaluations of text generation focused on fundamental skills, including similarity measures such as ROUGE (Lin, 2004) and BLEU (Papineni et al., 2002) for machine translation. They also included inherent scores like Diversity (Li et al., 2016) and Perplexity (Jelinek et al., 1977) for open-domain conversations (Zhang et al., 2018; Li et al., 2017), and factual measures such as F-score and exact match for machine reading comprehension (Rajpurkar et al., 2016; Chen, 2018) and question answering (Zhuang et al., 2023; Kwiatkowski et al., 2019). Human preference remains a critical aspect of model evaluation (Zheng et al., 2023b; Li et al., 2023b; Chiang et al., 2024). ChatbotArena (Chiang et al., 2024) employs a pairwise comparison approach and leverages crowd-sourcing to ensure task diversity.

**Neural Algorithmic Reasoning.** The creation of neural networks capable of executing algorithmic computations offers a promising bridge between classical algorithms and real-world data formats that defy traditional processing (Veličković & Blundell, 2021; Bevilacqua et al., 2023). Such networks further hold promise as replacements for human-crafted heuristics in combinatorial optimization, potentially surpassing their efficiency or effectiveness (Bengio et al., 2021; Georgiev et al., 2024). This field has evolved through multiple methodological approaches: graph neural networks (Khalil et al., 2017; Veličković et al., 2020), reinforcement learning frameworks (Mazyavkina et al., 2021), and LLMs (McLeish et al., 2024; Taylor et al., 2024; Fu et al., 2023), as well as emerging architectures combining GNNs and LLMs (Bounsi et al., 2024; Perozzi et al., 2024). Current evaluation paradigms emphasize distinct reasoning dimensions: While benchmarks like CLRS (Veličković et al., 2022) and its text-based adaptation (Markeeva et al., 2024) assess broad algorithmic mastery across diverse tasks, GraphArena specifically probes the depth of reasoning in graph-structured computations.

**LLM on Graphs.** Compared with graph learning tasks such as node classification (Chen et al., 2024c; Tang et al., 2023a) and link prediction (Liu et al., 2024; Tang et al., 2023b), graph computational problems pose more challenges for LLMs as they require a deeper understanding of structural information and long-range multi-step reasoning (Luo et al., 2024; Wang et al., 2024; Chai et al., 2023; Liu & Wu, 2023; Zhang et al., 2024b). GraphQA (Fatemi et al., 2023) and NLGraph (Wang et al., 2023) are two early benchmarks in this domain, focusing on basic graph problems using small-scale synthetic graphs. VisionGraph (Li et al., 2024) and GITA (Wei et al., 2024) introduce multimodal graph reasoning tasks that require LLMs to reason over both text and image modalities. GraphWiz (Chen et al., 2024a) introduces an instruction-tuning dataset to improve LLMs' graph reasoning capabilities.

## 5 CONCLUSION

This paper introduces GraphArena, a comprehensive benchmarking tool designed to assess the reasoning capabilities of LLMs on graph computational problems. Featuring a realistic graph collection, carefully curated tasks, and a rigorous evaluation framework, GraphArena provides valuable insights into LLM assessment and further development. Our extensive evaluation of ten leading LLMs across 10,000 graph computational problems reveals persistent hallucination issues, particularly evident with larger graphs, complex tasks, and models with fewer parameters. To address these challenges, we investigated four potential solutions: chain-of-thought prompting, instruction tuning, code writing, and scaling test-time compute. Each approach demonstrated unique strengths and limitations in mitigating model hallucination and improving LLM performance on graph problem-solving. Our findings suggest promising directions for future research, such as (1) exploring advanced finetuning techniques to enhance LLMs' code writing and algorithmic reasoning abilities, (2) developing more efficient approaches to scale test-time compute and address hallucination, and (3) integrating external tools or encoder modules to better support LLMs in graph computation.

**Acknowledgement.** This work was supported by National Key Research and Development Program of China Grant No. 2023YFF0725100 and Guangzhou-HKUST(GZ) Joint Funding Scheme 2023A03J0673.

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

## A  ADDITIONAL DATASET INFORMATION

As introduced in Section 2.1, problems in GraphArena is collected from five resources:

- **DBLP (Ley, 2002)** is an academic database containing over 7 million publications under the CC0 1.0 license. We use an undirected research collaboration graph for the period 2003-2018 from DBLP (Tang et al., 2008). This graph consists of 1,354,852 nodes and 5,129,047 edges, with each node representing an author and each edge reflecting a collaboration that has occurred at least five times.
- **Social Network (Rossi & Ahmed, 2015)** is a repository comprising over 70 different social networks under the CC BY-SA 3.0 License. We combine graphs from four large networks—Digg, Flixster, Lastfm, and Pokec—to form a dataset of 6,118,793 nodes and 40,647,227 edges. Each node symbolizes a user, and each edge denotes a friendship connection. All user names are pseudonymized for privacy.
- **DBpedia (Bizer et al., 2009)** extracts structured content from Wikipedia to form a comprehensive knowledge graph. We use DBpedia1M (Xin et al., 2022), a subset of the English version, which comprises 1,160,306 nodes and 7,214,631 edges. Each node represents an entity, and each edge denotes a relationship, with all original textual content retained as attributes. It is under the CC BY-SA 3.0 License.
- **Openflights (OpenFlights)** provides data for over 10,000 airports and their connecting flight routes. We construct an undirected graph where nodes are airports and edges are flight routes, weighted by the geographical distance between airports. We preserve the largest connected component in this graph, which consists of 3,390 nodes and 19,166 edges. Subsequently, we transform it into a fully-connected graph by adding new edges for all node pairs, using the shortest distance as the weight. This dataset is available under the DbCL v1.0 License.
- **PubChemQC (Nakata & Shimazaki, 2017)** offers a quantum chemistry database detailing ground-state electronic structures. From the PCQM4Mv2 Dataset—part of the OGB Large-Scale Challenge (Hu et al., 2021) collected from PubChemQC—we incorporate 3,746,620 individual graphs. Each graph represents a molecule, where nodes are atoms and edges are chemical bonds, with atom types preserved as node attributes. It is under the CC BY 4.0 License.

## B  ADDITIONAL TASK INFORMATION

We complete the description of remaining NP-complete tasks not fully detailed in Section 2.2:

- **Maximum Independent Set (MIS):** Select the largest set of mutually nonadjacent nodes from a graph $\mathcal{G} = \{\mathcal{V}, \mathcal{E}\}$. The task is to (1) identify an independent set $\mathcal{S} \subseteq \mathcal{V}$, and (2) ensure that $\mathcal{S}$ is the largest among all feasible sets. This task is performed using the Social Network dataset.
- **Minimum Vertex Cover (MVC):** Given a graph $\mathcal{G} = \{\mathcal{V}, \mathcal{E}\}$, find a subset of nodes $\mathcal{S} \subseteq \mathcal{V}$ such that each edge in $\mathcal{G}$ has at least one endpoint in $\mathcal{S}$. The Minimum Vertex Cover Problem is to (1) find the vertex cover $\mathcal{S}$ and (2) ensure the size of $\mathcal{S}$ is minimum among all vertex covers of $\mathcal{G}$. We explore this using the Social Network dataset.
- **Maximum Common Subgraph (MCS):** Compare two graphs $\mathcal{G}$ and $\mathcal{H}$ to find the largest common subgraph. The objectives are to (1) determine a node-induced common subgraph $\mathcal{S}$ between $\mathcal{G}$ and $\mathcal{H}$, and (2) maximize the size of $\mathcal{S}$. The PubChemQC dataset is used for this task.
- **Graph Edit Distance (GED):** For two graphs $\mathcal{G}$ and $\mathcal{H}$, determine the minimum edit distance via node mappings. This involves (1) establishing a node mapping that aligns $\mathcal{G}$ with $\mathcal{H}$, and (2) minimizing the edit operations required. These operations include adding or deleting an edge or isolated node, or relabeling a node. The PubChemQC dataset is utilized here.
- **Traveling Salesman Problem (TSP):** Solve the TSP in a weighted graph $\mathcal{G} = \{\mathcal{V}, \mathcal{E}\}$, where $w : \mathcal{E} \to \mathbb{R}^+$ assigns a positive weight to each edge, representing the travel distance. The objectives are to (1) find a route $\mathcal{P}$ that visits each node exactly once and returns to the starting point, and (2) minimize the total travel distance. We utilize the OpenFlights dataset for this challenge. To ensure every pair of nodes is connected, thereby guaranteeing a feasible solution, we convert the dataset into a complete graph by adding edges to represent the shortest possible distances between nodes that are not directly connected.

Table 4 presents comprehensive statistics for each task in GraphArena, including the average and maximum number of nodes, edges, and text length (in characters). Tables 9-18 provide detailed examples of prompts for each task. For the Connected Component and Graph Diameter tasks, chain-of-thought prompting with step-by-step solution demonstrations is provided.

## C  ADDITIONAL EXPERIMENTAL RESULTS

Figure 8 offers a feasibility and accuracy comparison of the remaining five LLMs on individual tasks, complementing the analysis in Figure 2. Figures 9 and 10 illustrate the influence of graph size on hallucination probability for the remaining seven tasks, complementing Figure 4. Table 5 extends the analysis in Figure 3, showing the percentage of problems where each of the 10 LLMs wins, ties, or loses against Random, Greedy, and Approximated graph algorithms.

**Comparison with graph neural networks (GNNs).** Directly comparing GNNs and LLMs in GraphArena is challenging due to their fundamentally different paradigms. LLMs are multi-task, unsupervised models that process natural language inputs, while GNNs are typically task-specific, supervised models that operate on network data. For simplicity, we configured GNNs to directly predict the answer rather than generating the solution path or component. Although it is feasible to let GNNs generate detailed solutions, different tasks like GED and MCS would require specific architectural modifications. We compared three representative GNNs: Graph Isomorphism Network (GIN) (Xu et al., 2019), Graph Attention Network (GAT) (Veličković et al., 2017), and Graph Sample and Aggregate (GSAGE) (Hamilton et al., 2017). We generated an equivalent amount of training data for GNNs using the same framework as GraphArena. Table 6 shows the average accuracy for these GNNs on Polynomial-time and NP-complete problems across small and large graphs. The results indicate that Claude-3.5-sonnet demonstrates superior performance in most scenarios. While GNNs achieve the best results on large-scale NP-complete problems, recent studies have theoretically proved the limitations in GNNs' computational capabilities (Zhang et al., 2024a). For instance, message passing neural networks cannot even count triangles. Given the limitations, we posit that GNNs might be performing pattern-based regression rather than genuinely solving graph problems. In contrast, LLMs appear more aligned with true problem-solving.

**Comparison of different graph tokenizers.** In Table 7, we examined the performance of three graph tokenizers—Edge List, Adjacency List, and Adjacency Matrix—on GPT-4o across two selected tasks: Shortest Distance and TSP, both on small and large graphs. As shown in The Edge List tokenizer demonstrated relatively stable performance across various tasks, with feasibility and accuracy scores consistently higher than those of the Adjacency Matrix, particularly in the Shortest Distance tasks where it achieved 0.914 and 0.902 accuracy for small graphs, and 0.788 and 0.720 for large graphs. Although the Adjacency List tokenizer slightly outperformed the Edge List in some instances, such as in the TSP (Large) task with a lower hallucination rate of 0.082, the Edge List remains the most stable option. Combined with its widespread use in prior studies like NLGraph (Wang et al., 2023) and GraphWiz (Chen et al., 2024a), we use Edge List as our primary graph tokenizer for the main experiments, as shown in Tables 9 to 18. Conversely, the Adjacency Matrix often introduced redundant information, particularly in sparse graphs, leading to decreased performance, as evidenced by its 0.436 hallucination rate and 0.448 accuracy in the Shortest Distance (Large) task.

**Comparison of Real-World and Synthetic Graphs.** Table 8 presents the evaluation of GPT-4o on two polynomial-time tasks: Common Neighbor and Shortest Distance. Our findings indicate that GPT-4o achieves notably higher accuracy and lower hallucination rates on synthetic Erdős-Rényi graphs compared to real-world graphs of similar size. This disparity in performance suggests that synthetic graphs may fail to adequately represent the complex topological features and irregular structures found in real-world networks. Consequently, relying exclusively on synthetic benchmarks could lead to an overestimation of a model's generalization capabilities. This observation motivates us to incorporate samples from real-world networks into GraphArena.

Table 4: Summary of task statistics in GraphArena, detailing the average and maximum values for nodes ($V$), edges ($E$), text length ($T$) in characters, graph density ($D$), and edge connectivity ($EC$) across problem instances.

| Problems | $\overline{|V|}$ | $|V|_{max}$ | $\overline{|E|}$ | $|E|_{max}$ | $\overline{|T|}$ | $|T|_{max}$ | $\overline{D}$ | $\overline{EC}$ |
|---|---|---|---|---|---|---|---|---|
| Common Neighbor | 19.4 | 50 | 68.6 | 409 | 4323.5 | 18014 | 0.44 | 1.37 |
| Shortest Distance | 19.6 | 50 | 36.8 | 209 | 5639.5 | 23731 | 0.28 | 0.61 |
| Connected Component | 13.5 | 30 | 14.4 | 122 | 1785.4 | 5503 | 0.23 | 0.26 |
| Graph Diameter | 13.4 | 30 | 22.0 | 128 | 3741.3 | 11783 | 0.31 | 1.16 |
| Maximum Clique Problem | 13.2 | 30 | 39.1 | 154 | 3025.0 | 7921 | 0.52 | 1.70 |
| Maximum Independent Set | 12.8 | 30 | 19.5 | 141 | 1902.8 | 6182 | 0.30 | 1.08 |
| Minimum Vertex Cover | 14.1 | 30 | 23.3 | 151 | 2178.4 | 6429 | 0.31 | 1.17 |
| Maximum Common Subgraph | 9.2 | 20 | 8.9 | 22 | 1025.1 | 1187 | 0.30 | 1.04 |
| Graph Edit Distance | 9.4 | 20 | 9.2 | 23 | 1643.7 | 2142 | 0.29 | 1.03 |
| Traveling Salesman Problem | 9.5 | 20 | 50.6 | 190 | 1806.1 | 4741 | 1.00 | 8.46 |

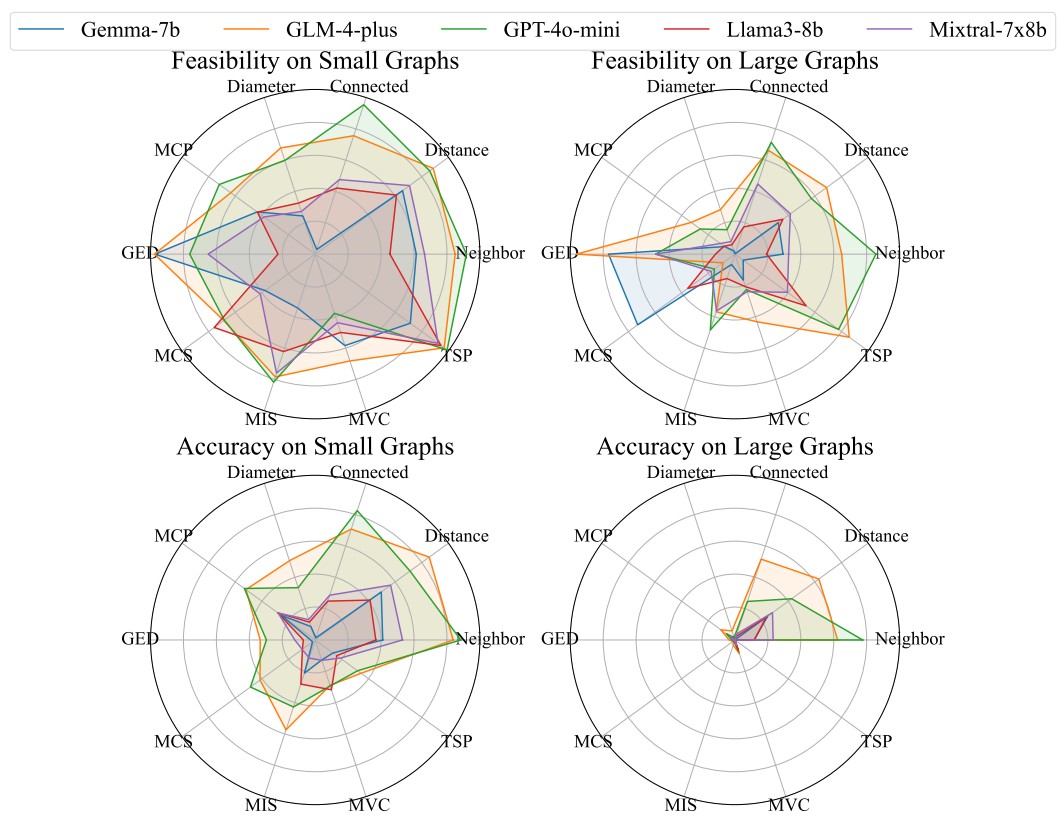

Figure 8: Feasibility and accuracy comparison of the remaining five LLMs in Figure 2 on each individual task.

Table 5: The percentage of problems where **LLM wins, ties, or loses against three types of classic graph algorithms**: Random, Greedy, and Approximated.

| | Travelling Salesman Problem (Small) | | | Travelling Salesman Problem (Large) | | |
|---|---|---|---|---|---|---|
| Algorithm | Random | Greedy | Approximated | Random | Greedy | Approximated |
| LLM | win/tie/lose | win/tie/lose | win/tie/lose | win/tie/lose | win/tie/lose | win/tie/lose |
| GPT-4o | 76.2/15.2/8.6 | 23.6/49.0/27.4 | **18.8**/36.2/45.0 | 95.2/0.0/4.8 | 18.8/4.2/77.0 | 10.6/0.6/88.8 |
| GPT-4o-mini | 67.6/14.8/17.6 | 17.2/40.4/42.4 | 12.4/29.4/58.2 | 70.6/0.0/29.4 | 0.6/0.0/99.4 | 0.0/0.0/100.0 |
| Claude-3.5-sonnet | **77.4**/15.8/6.8 | **24.6**/52.2/23.2 | 17.4/40.6/42.0 | **99.6**/0.0/0.4 | **33.2**/8.4/58.4 | **20.8**/1.0/78.2 |
| Llama3-70b | 65.0/13.4/21.6 | 12.0/36.6/51.4 | 8.4/23.8/67.8 | 78.6/0.0/21.4 | 1.4/0.0/98.6 | 0.6/0.0/99.4 |
| Llama3-8b | 47.0/13.6/39.4 | 12.2/15.4/72.4 | 6.4/15.6/78.0 | 30.6/0.0/69.4 | 0.2/0.0/99.8 | 0.0/0.0/100.0 |
| Qwen2.5-72b | 45.6/11.6/42.8 | 14.6/19.0/66.4 | 9.0/19.6/71.4 | 30.0/0.0/70.0 | 1.0/0.4/98.6 | 0.8/0.0/99.2 |
| Deepseek-v2 | 72.6/15.4/12.0 | 21.0/42.0/37.0 | 15.8/32.2/52.0 | 66.8/0.0/33.2 | 4.4/0.2/95.4 | 2.2/0.0/97.8 |
| Mixtral-7x8b | 46.4/14.2/39.4 | 10.6/22.6/66.8 | 6.0/18.0/76.0 | 24.4/0.0/75.6 | 0.0/0.0/100.0 | 0.0/0.0/100.0 |
| Gemma-7b | 30.2/12.6/57.2 | 8.8/13.8/77.4 | 5.0/13.0/82.0 | 3.4/0.0/96.6 | 0.0/0.0/100.0 | 0.0/0.0/100.0 |
| GLM-4-plus | 69.8/14.0/16.2 | 19.6/39.4/41.0 | 13.2/31.0/55.8 | 84.0/0.0/16.0 | 5.2/0.2/94.6 | 2.6/0.4/97.0 |
| | Graph Edit Distance (Small) | | | Graph Edit Distance (Large) | | |
| GPT-4o | 71.0/14.6/14.4 | 35.0/49.2/15.8 | 12.4/30.8/56.8 | **94.6**/1.4/4.0 | 49.6/18.2/32.2 | 34.2/3.8/62.0 |
| GPT-4o-mini | 57.2/12.2/30.6 | 30.2/37.8/32.0 | 10.6/28.4/61.0 | 47.0/0.8/52.2 | 29.4/10.2/60.4 | 25.4/3.0/71.6 |
| Claude-3.5-sonnet | **72.2**/12.6/15.2 | 41.0/43.2/15.8 | 13.0/33.6/53.4 | 82.4/0.8/16.8 | **52.4**/13.2/34.4 | 35.6/5.4/59.0 |
| Llama3-70b | 69.6/14.8/15.6 | **41.2**/30.0/28.8 | 10.6/32.0/57.4 | 80.0/0.8/19.2 | 41.8/10.6/47.6 | 28.4/3.4/68.2 |
| Llama3-8b | 14.2/5.2/80.6 | 10.8/6.2/83.0 | 3.0/7.2/89.8 | 11.8/0.0/88.2 | 10.8/0.4/88.8 | 6.6/1.0/92.4 |
| Qwen2.5-72b | 64.0/10.0/26.0 | 37.4/33.2/29.4 | 11.2/34.6/54.2 | 20.8/0.2/79.0 | 14.0/3.2/82.8 | 10.8/1.8/87.4 |
| Deepseek-v2 | 61.6/14.0/24.4 | 35.4/32.8/31.8 | 10.6/26.8/62.6 | 82.4/1.2/16.4 | 45.6/13.2/41.2 | 32.6/4.4/63.0 |
| Mixtral-7x8b | 37.8/12.4/49.8 | 17.6/21.4/61.0 | 6.0/11.4/82.6 | 43.6/1.2/55.2 | 15.8/4.4/79.8 | 9.4/1.2/89.4 |
| Gemma-7b | 53.6/22.4/24.0 | 30.8/21.0/48.2 | 3.6/6.4/90.0 | 75.4/1.2/23.4 | 43.0/7.2/49.8 | 25.0/3.4/71.6 |
| GLM-4-plus | **72.2**/16.0/11.8 | 38.0/46.8/15.2 | **14.2**/28.6/57.2 | 94.0/0.8/5.2 | 52.0/21.6/26.4 | **37.6**/3.8/58.6 |
| | Maximum Independent Set (Small) | | | Maximum Independent Set (Large) | | |
| GPT-4o | **67.6**/12.4/20.0 | 0.0/63.8/36.2 | **2.2**/64.8/33.0 | 29.8/1.0/69.2 | 0.0/5.4/94.6 | 0.4/8.2/91.4 |
| GPT-4o-mini | 63.8/11.0/25.2 | 0.0/42.8/57.2 | 0.8/44.6/54.6 | 40.4/3.8/55.8 | 0.0/1.8/98.2 | 0.0/2.4/97.6 |
| Claude-3.5-sonnet | 61.8/11.4/26.8 | 0.0/68.0/32.0 | 1.8/68.0/30.2 | 25.2/1.0/73.8 | 0.0/7.8/92.2 | **0.8**/8.8/90.4 |
| Llama3-70b | 62.4/21.4/16.2 | 0.0/36.8/63.2 | 0.6/38.2/61.2 | **42.6**/3.4/54.0 | **0.2**/2.6/97.2 | 0.6/3.8/95.6 |
| Llama3-8b | 46.0/10.0/44.0 | **0.4**/28.0/71.6 | 1.4/29.2/69.4 | 14.2/0.4/85.4 | 0.0/1.0/99.0 | 0.2/1.4/98.4 |
| Qwen2.5-72b | 52.0/21.8/26.2 | 0.0/3.8/96.2 | 0.6/5.0/94.4 | 39.0/4.2/56.8 | 0.0/1.0/99.0 | 0.4/1.8/97.8 |
| Deepseek-v2 | 55.4/15.6/29.0 | 0.0/36.2/63.8 | 0.8/37.4/61.8 | 12.2/0.6/87.2 | 0.0/3.2/96.8 | 0.4/3.6/96.0 |
| Mixtral-7x8b | 44.0/16.2/39.8 | 0.0/11.6/88.4 | 0.4/12.6/87.0 | 28.4/2.8/68.8 | **0.2**/0.8/99.0 | 0.6/0.6/98.8 |
| Gemma-7b | 26.6/5.6/67.8 | 0.0/21.2/78.8 | 0.0/21.2/78.8 | 6.0/0.0/94.0 | 0.0/0.6/99.4 | 0.0/0.6/99.4 |
| GLM-4-plus | 64.2/11.6/24.2 | 0.2/57.2/42.6 | **2.2**/58.8/39.0 | 34.4/0.4/65.2 | 0.0/3.8/96.2 | 0.6/5.4/94.0 |
| | Minimum Vertex Cover (Small) | | | Minimum Vertex Cover (Large) | | |
| GPT-4o | 40.6/7.6/51.8 | 39.8/7.0/53.2 | 30.0/16.4/53.6 | 19.0/1.2/79.8 | 13.6/2.6/83.8 | 12.0/3.8/84.2 |
| GPT-4o-mini | 34.6/3.2/62.2 | 27.4/8.2/64.4 | 25.4/9.6/65.0 | 21.6/0.2/78.2 | 11.8/3.6/84.6 | 11.4/6.2/82.4 |
| Claude-3.5-sonnet | 16.6/1.2/82.2 | 16.6/1.2/82.2 | 12.8/5.0/82.2 | 8.6/0.0/91.4 | 8.0/0.6/91.4 | 7.4/1.0/91.6 |
| Llama3-70b | 41.0/6.0/53.0 | **40.0**/7.4/52.6 | **32.0**/14.4/53.6 | 17.0/0.0/83.0 | 12.2/4.4/83.4 | 12.0/3.8/84.2 |
| Llama3-8b | 38.8/9.6/51.6 | 32.4/11.8/55.8 | 27.0/15.8/57.2 | 17.6/0.0/82.4 | 10.2/2.0/87.8 | 9.4/4.2/86.4 |
| Qwen2.5-72b | 2.8/0.8/96.4 | 3.4/0.2/96.4 | 2.0/1.6/96.4 | 1.6/0.0/98.4 | 0.6/0.0/99.4 | 0.6/0.0/99.4 |
| Deepseek-v2 | 45.8/9.2/45.0 | 36.6/14.8/48.6 | 31.6/17.0/51.4 | 29.0/1.8/69.2 | 10.6/8.4/81.0 | 11.2/8.8/80.0 |
| Mixtral-7x8b | 28.2/11.4/60.4 | 15.4/20.4/64.2 | 12.6/19.0/68.4 | 21.2/0.8/78.0 | 6.2/3.8/90.0 | 5.6/4.4/90.0 |
| Gemma-7b | 38.2/14.2/47.6 | 15.0/30.4/54.6 | 15.0/25.6/59.4 | 15.8/0.2/84.0 | 4.8/2.4/92.8 | 3.6/4.4/92.0 |
| GLM-4-plus | **53.0**/10.4/36.6 | 31.2/20.4/48.4 | 29.0/23.6/47.4 | **33.8**/1.6/64.6 | **18.4**/7.2/74.4 | **15.4**/9.4/75.2 |

Table 6: Average accuracy (%) for three representative GNNs on **P**olynomial-time and **NP**-complete problems across small and large graphs.

| Models | P (Small) | P (Large) | NP (Small) | NP (Large) |
|---|---|---|---|---|
| GIN | 59.9 | 39.6 | 33.8 | **25.8** |
| GAT | 42.3 | 22.5 | 32.3 | 17.9 |
| GSAGE | 41.0 | 22.6 | 33.7 | 17.6 |
| Claude-3.5-sonnet | **82.2** | **58.7** | **47.8** | 7.2 |

Table 7: Performance comparison for different graph tokenizers on GPT-4o across tasks.

| Tasks | Distance (Small) | | | Distance (Large) | | | TSP (Small) | | | TSP (Large) | | |
|---|---|---|---|---|---|---|---|---|---|---|---|---|
| Graph Tokenizer | Fea. | Hallu. | Acc. | Fea. | Hallu. | Acc. | Fea. | Hallu. | Acc. | Fea. | Hallu. | Acc. |
| Edge List | **0.914** | **0.086** | **0.902** | 0.788 | 0.192 | 0.720 | **0.994** | **0.004** | **0.522** | 0.870 | 0.128 | 0.000 |
| Adjacency List | 0.898 | 0.102 | 0.894 | **0.808** | **0.166** | **0.774** | 0.990 | 0.006 | 0.288 | **0.918** | 0.082 | 0.000 |
| Adjacency Matrix | 0.842 | 0.158 | 0.780 | 0.546 | 0.436 | 0.448 | 0.940 | 0.058 | 0.318 | 0.806 | **0.012** | **0.016** |

Table 8: Performance comparison for GPT-4o on real-world and synthetic graphs.

| Tasks | Neighbor (Small) | | | Neighbor (Large) | | | Distance (Small) | | | Distance (Large) | | |
|---|---|---|---|---|---|---|---|---|---|---|---|---|
| Graph Type | Fea. | Hallu. | Acc. | Fea. | Hallu. | Acc. | Fea. | Hallu. | Acc. | Fea. | Hallu. | Acc. |
| Real-world Graphs | 0.838 | 0.088 | 0.778 | 0.740 | 0.140 | 0.626 | 0.922 | 0.078 | 0.904 | 0.760 | 0.224 | 0.680 |
| Erdős-Rényi Graphs | **0.996** | **0.004** | **0.986** | **0.984** | **0.016** | **0.730** | **0.980** | **0.020** | **0.946** | **0.864** | **0.146** | **0.702** |

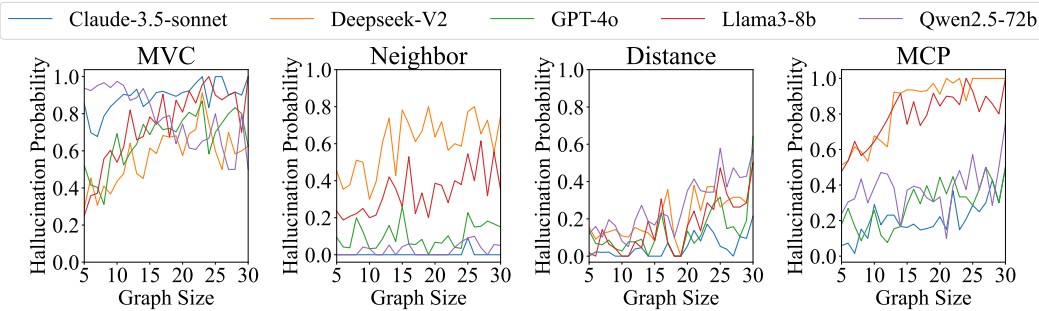

Figure 9: The influence of graph size on hallucination probability for the tasks of Distance, Neighbor, MCP, and MVC.

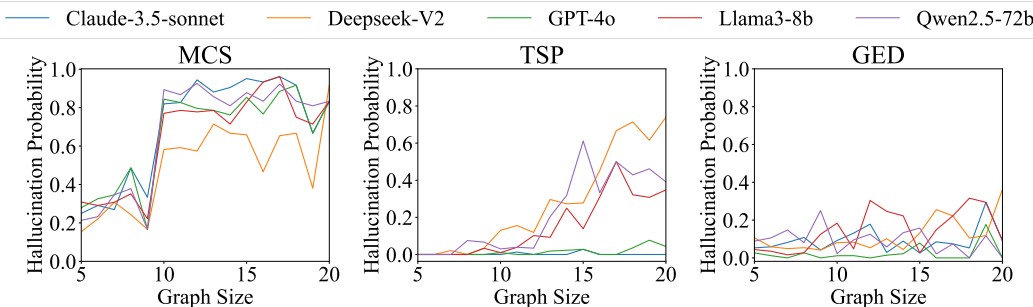

Figure 10: The influence of graph size on hallucination probability for the tasks of GED, TSP, and MCS.

Table 9: An example of CoT prompting with step-by-step demonstrations on the Component task.

**CoT Prompts of the Connected Component Task. (1-shot)**

You are required to identify all connected components in the given social network and output one representative node from each component.
Within a connected component, any node can be reached from any other node through the edges in the graph. Different connected components are isolated from each other.

**Example**

- Names in the network: Michelle Nelson, Kevin Brooks, Eric Davis, Toni Suarez, Teresa Hughes
- Friendship connections: Michelle Nelson to Teresa Hughes, Kevin Brooks to Eric Davis, Eric Davis to Toni Suarez
The answer includes one representative element from each connected component in the given social network: In this social network, we have the following names and connections: Michelle Nelson, Kevin Brooks, Eric Davis, Toni Suarez, and Teresa Hughes. The connections are: Michelle Nelson to Teresa Hughes, Kevin Brooks to Eric Davis, and Eric Davis to Toni Suarez. From these connections, we can identify two isolated connected components. The first component includes Michelle Nelson and Teresa Hughes, and the second component includes Kevin Brooks, Eric Davis, and Toni Suarez. By selecting one representative node from each connected component, we have Teresa Hughes from the first component and Kevin Brooks from the second component. Therefore, the answer is [Teresa Hughes, Kevin Brooks].

**Problem to Solve**

- Names in the network: Michelle Nelson, Kevin Brooks, Eric Davis, Toni Suarez, Teresa Hughes
- Friendship connections: Michelle Nelson to Teresa Hughes, Kevin Brooks to Eric Davis, Eric Davis to Toni Suarez

Identify all connected components in this network. Note that for each connected component, you should only output one of its nodes. Present your answer in the following format: [UserA, UserB, UserC, UserD, ...]

Table 10: An example of CoT prompting with step-by-step demonstrations on the Diameter task.

**CoT Prompts of the Graph Diameter Task. (1-shot)**

You are required to calculate the diameter of an undirected knowledge graph.
The diameter of a graph is the maximum distance between any pair of nodes in the graph. To compute this, you need to find the shortest path between all pairs of nodes and then determine the maximum length of these shortest paths.

**Example**

-Entities in this knowledge graph: SeineetMarne, MontereauFaultYonne, Vittorio De Sica, SaintFiacre SeineetMarne, France
- The relationships between these entities are as follows: SeineetMarne is connected to SaintFiacre SeineetMarne via the relationship department. SeineetMarne is connected to MontereauFaultYonne via the relationship department. SeineetMarne is connected to France via the relationship country. MontereauFaultYonne is connected to France via the relationship country. Vittorio De Sica is connected to France via the relationship deathPlace. SaintFiacre SeineetMarne is connected to France via the relationship country.

To calculate the diameter of this undirected knowledge graph, we first need to understand the connections and the structure of the graph. The entities and their relationships are as follows: SeineetMarne is connected to SaintFiacre SeineetMarne via the relationship 'department', SeineetMarne is connected to MontereauFaultYonne via the relationship 'department', SeineetMarne is connected to France via the relationship 'country', MontereauFaultYonne is connected to France via the relationship 'country', Vittorio De Sica is connected to France via the relationship 'deathPlace', and SaintFiacre SeineetMarne is connected to France via the relationship 'country'. The graph structure can be visualized as follows: SeineetMarne is connected to SaintFiacre SeineetMarne, MontereauFaultYonne, and France. MontereauFaultYonne and SaintFiacre SeineetMarne are also connected to France, and France is connected to Vittorio De Sica. By analyzing the shortest paths between all pairs of nodes, we find that SeineetMarne to SaintFiacre SeineetMarne, SeineetMarne to MontereauFaultYonne, SeineetMarne to France, MontereauFaultYonne to France, SaintFiacre SeineetMarne to France, and France to Vittorio De Sica all have a direct connection with 1 edge. The longest shortest paths are from SeineetMarne, MontereauFaultYonne, and SaintFiacre SeineetMarne to Vittorio De Sica via France, each with 2 edges. Therefore, the diameter of this knowledge graph is 2, and one of the paths that represent this diameter is [SeineetMarne, France, Vittorio De Sica].

**Problem to Solve**

- Entities in this knowledge graph: PlayStation 2, Buzz! Junior: Jungle Party, Tekken (video game), Buzz!, PlayStation Network
- The relationships between these entities are as follows:
PlayStation 2 is connected to Tekken (video game) via the relationship computingPlatform. PlayStation 2 is connected to Buzz! Junior: Jungle Party via the relationship computingPlatform. Buzz! Junior: Jungle Party is connected to Buzz! via the relationship series. Buzz! Junior: Jungle Party is connected to PlayStation Network via the relationship computingPlatform. Tekken (video game) is connected to PlayStation Network via the relationship computingPlatform.

Please determine the diameter of this network and output the corresponding path in the following format: [Entity1, Entity2, ..., EntityN].

Table 11: An example of the Common Neighbor task.

**Prompts of the Common Neighbor Task. (1-shot)**

Your task is to find the common neighbors of two nodes in an undirected academic network. In this network, nodes represent authors and edges represent research collaborations.

**Example**

- Authors in the network: Marie-Francine Moens, Stefanie Brüninghaus, Henry Prakken, David W. Aha, Kevin D. Ashley, Ronald Prescott Loui
- Research collaborations between these authors: Marie-Francine Moens and Kevin D. Ashley, Marie-Francine Moens and Stefanie Brüninghaus, Stefanie Brüninghaus and David W. Aha, Stefanie Brüninghaus and Henry Prakken, Stefanie Brüninghaus and Kevin D. Ashley, Stefanie Brüninghaus and Ronald Prescott Loui, Henry Prakken and Kevin D. Ashley, Henry Prakken and Ronald Prescott Loui, David W. Aha and Kevin D. Ashley, Kevin D. Ashley and Ronald Prescott Loui.
Common neighbors between Marie-Francine Moens and Stefanie Brüninghaus: [Kevin D. Ashley]
**Problem to Solve**

- Authors in the network: Sang Wan Lee, Hyoyoung Jang, Z. Zenn Bien, Zeungnam Bien
- Research collaborations between these authors: Sang Wan Lee and Zeungnam Bien, Sang Wan Lee and Z. Zenn Bien, Sang Wan Lee and Hyoyoung Jang, Hyoyoung Jang and Zeungnam Bien, Z. Zenn Bien and Zeungnam Bien.

Please identify the common neighbors of Sang Wan Lee and Hyoyoung Jang in this network. Present your answer in the following format: [AuthorA, AuthorB, AuthorC, AuthorD, ...].

Table 12: An example of the Shortest Distance task.

**Prompts of the Shortest Distance Task. (1-shot)**

Your task is to identify the shortest path between two specified entities in an undirected knowledge graph, minimizing the number of hops.

**Example**

- Entities in this knowledge graph: Chordata, Animalia, Sloggett's vlei rat, Mammalia
- The relationships between these entities are as follows:
Chordata is connected to Sloggett's vlei rat via the relationship phylum. Animalia is connected to Sloggett's vlei rat via the relationship kingdom. Sloggett's vlei rat is connected to Mammalia via the relationship class.
One shortest path between Animalia and Sloggett's vlei rat is: [Animalia, Sloggett's vlei rat]

**Problem to Solve**

- Entities in this knowledge graph: United States, Post-hardcore, Head Wound City, Texas, Cody Votolato
- The relationships between these entities are as follows:
United States is connected to Texas via the relationship country. United States is connected to Cody Votolato via the relationship birthPlace. Post-hardcore is connected to Cody Votolato via the relationship genre. Head Wound City is connected to Cody Votolato via the relationship associatedMusicalArtist. Texas is connected to Cody Votolato via the relationship birthplace.

Please determine the shortest path between Texas and United States in this network. Submit your answer in the format: [Entity1, Entity2, ..., EntityN], where Entity1 and EntityN are the specified start and end entities, and Entity2 through EntityN-1 are the intermediate entities on the shortest path.

Table 13: An example of the Graph Edit Distance task.

**Prompts of the GED Task. (1-shot)**

You are required to solve the Graph Edit Distance problem between two molecules. Each edit operation (adding or deleting an edge, adding or deleting an isolated node, or relabeling a node) has an identity cost. Your objective is to establish a mapping between the atom IDs from Molecule A to Molecule B, ensuring that each atom ID in Molecule A corresponds to exactly one atom ID in Molecule B. The mapping corresponds to the minimum edit cost between the two graphs.

**Example**

Molecule A:
- Atoms: N (atom 0), O (atom 1), Si (atom 2), O (atom 3), O (atom 4).
- Bonds: 0-1, 1-2, 2-3, 2-4.
Molecule B:
- Atoms: F (atom 0), B (atom 1), C (atom 2), N (atom 3), Br (atom 4).
- Bonds: 0-1, 1-2, 1-4, 2-3.
One optimal node mapping: [3, 2, 1, 0, 4].

**Problem to Solve**

You are given the following two molecules:

Molecule A:
Atoms: N (atom 0), C (atom 1), N (atom 2), F (atom 3).
Bonds: 0-1, 1-2, 1-3.
Molecule B:
Atoms: O (atom 0), C (atom 1), F (atom 2), F (atom 3).
Bonds: 0-1, 1-2, 1-3.

Represent the node mapping as a list of integers, where the position in the list corresponds to the atom ID in Molecule A and the value at that position indicates the corresponding atom ID in Molecule B.

For instance, if atom 0 in Molecule A corresponds to atom 1 in Molecule B, atom 1 in Molecule A corresponds to atom 0 in Molecule B, and atom 2 remains unchanged, the mapping would be represented as [1, 0, 2, ...].

Table 14: An example of the Maximum Clique Problem.

**Prompts of the MCP Task. (1-shot)**

You are required to solve the Maximum Clique Problem for an undirected academic network. In this network, nodes represent authors and edges represent research collaborations. Your objective is to find the largest subset of nodes such that every pair of vertices in this subset is connected by an edge.

**Example**

- Authors in the network: Keng Peng Tee, Veit Hagenmeyer, Bartosz Käpernick, Karl Henrik Johansson, Darryl DeHaan, James B. Rawlings, Andreas Kugi, Knut Graichen, Tilman Utz, Christian Ebenbauer.
- Research collaborations between these authors: Keng Peng Tee and James B. Rawlings, Keng Peng Tee and Darryl DeHaan, Veit Hagenmeyer and Knut Graichen, Bartosz Käpernick and Andreas Kugi, Bartosz Käpernick and Knut Graichen, Karl Henrik Johansson and Christian Ebenbauer, Darryl DeHaan and Knut Graichen, Darryl DeHaan and Christian Ebenbauer, James B. Rawlings and Knut Graichen, James B. Rawlings and Christian Ebenbauer, Andreas Kugi and Knut Graichen, Knut Graichen and Tilman Utz.
One Maximum Clique: [Knut Graichen, Bartosz Käpernick, Andreas Kugi].

**Problem to Solve**

- Authors in the network: Manfred Schmidt-Schauss, David Sabel, Manfred Schmidt-Schauß, Guillem Godoy.
- Research collaborations between these authors: Manfred Schmidt-Schauss and David Sabel, Manfred Schmidt-Schauss and Manfred Schmidt-Schauß, Manfred Schmidt-Schauss and Guillem Godoy, David Sabel and Manfred Schmidt-Schauß, Manfred Schmidt-Schauß and Guillem Godoy.

Identify the clique with the maximum number of authors in this network. Present your answer in the following format: [AuthorA, AuthorB, AuthorC, AuthorD, ...].

---

Table 15: An example of the Maximum Common Subgraph Task.

**Prompts of the MCS Task. (1-shot)**

You are required to solve the Maximum Common Subgraph problem. Your goal is to identify the common subgraph with the maximum number of atoms shared between the two molecules.

**Example**

Molecule A consists of 8 atoms with the following 9 bonds: 0-1, 0-6, 1-2, 2-3, 3-4, 3-7, 4-5, 5-6, 5-7.
Molecule B consists of 7 atoms with the following 7 bonds: 0-1, 1-2, 1-4, 2-3, 3-4, 3-6, 4-5.
One max common subgraph: [2, 3, 4, 5, 7, 6], [0, 1, 2, 3, 4, 6].

**Problem to Solve**

You are given the following two molecules:
Molecule A consists of 4 atoms with the following 3 bonds: 0-1, 1-2, 2-3. Molecule B consists of 4 atoms with the following 3 bonds: 0-1, 1-2, 1-3.
Provide the indices of the atoms in the common subgraph for each molecule in the following format: [Node indices in molecular A], [Node indices in molecular B].

For example, if the common subgraph is the subgraph of atom 1, 2, 3 in molecule A and the subgrah of atom 2, 3, 4 in molecule B, you should answer: [1, 2, 3], [2, 3, 4].

Table 16: An example of the Maximum Independent Set task.

**Prompts of the MIS Task. (1-shot)**

Your task is to solve the Maximum Independent Set problem in the given social network. In this network, each node represents a user, and each edge represents a friendship connection. You need to identify the largest subset of users such that no two users in this subset are friends connected by an edge.

**Example**

- Users in the network: Melinda Vaughan, Mary Thornton, Jeremiah Griffith, Lisa Anderson, Alfred Powell.
- Fiendship connections: Melinda Vaughan and Jeremiah Griffith, Mary Thornton and Jeremiah Griffith, Jeremiah Griffith and Lisa Anderson, Jeremiah Griffith and Alfred Powell.
One Maximum Independent Set: [Melinda Vaughan, Lisa Anderson, Alfred Powell, Mary Thornton].

**Problem to Solve**

- Users in the network: William Lawson, Daniel Shelton, Michelle Lewis, Julie Hayes.
- Friendship connections: William Lawson and Daniel Shelton, William Lawson and Julie Hayes, Michelle Lewis and Julie Hayes.

Identify the Maximum Independent Set of this network and present your answer in the following format: [UserA, UserB, UserC, UserD, ...].

Table 17: An example of the Minimum Vertex Cover task.

**Prompts of the MVC Task. (1-shot)**

Your task is to solve the Minimum Vertex Cover problem in the given social network. In this network, each node represents a user, and each edge represents a friendship connection. You need to identify the smallest subset of users such that every friendship connection has at least one user from this subset.

**Example**

- Users in the network: Julie Harris, David Torres, Vanessa Parker, Shawn Barnett, Karl Dean.
- Fiendship connections: Julie Harris and Vanessa Parker, Julie Harris and David Torres, Julie Harris and Shawn Barnett, Julie Harris and Karl Dean, David Torres and Vanessa Parker, David Torres and Shawn Barnett, David Torres and Karl Dean, Vanessa Parker and Shawn Barnett, Shawn Barnett and Karl Dean.
One Minimum Vertex Cover: [Julie Harris, David Torres, Shawn Barnett].

**Problem to Solve**

- Users in the network: Pamela Haynes, Kyle Meadows, Adam Nichols, Anna Lowery, Heather Dixon, Matthew Lee, Elizabeth Wood, Stephen Hess.
- Friendship connections: Pamela Haynes and Stephen Hess, Kyle Meadows and Matthew Lee, Kyle Meadows and Stephen Hess, Kyle Meadows and Adam Nichols, Adam Nichols and Stephen Hess, Adam Nichols and Heather Dixon, Anna Lowery and Stephen Hess, Heather Dixon and Stephen Hess, Matthew Lee and Stephen Hess, Elizabeth Wood and Stephen Hess.

Identify the Minimum Vertex Cover of this network and present your answer in the following format: [UserA, UserB, UserC, UserD, ...].

Table 18: An example of the Traveling Salesman Problem.

---

**Prompts of the TSP Task. (1-shot)**

---

You are required to solve the Traveling Salesman Problem for an undirected flight route network. Your objective is to determine the shortest possible route that visits each of the listed airports exactly once and returns to the starting point.

**Example**

- Airports to visit: VPY, AAE, BGA, YWB.
- Travel distances (in kilometers) between each pair of airports:
VPY to YWB: 16285; VPY to AAE: 9488; VPY to BGA: 13255; AAE to YWB: 7807; AAE to BGA: 9332; BGA to YWB: 6575.
One shortest route: [VPY, AAE, YWB, BGA, VPY].

**Problem to Solve**

- Airports to visit: YNT, TEB, CFC, VIE, NSH, ROV.
- Travel distances (in kilometers) between each pair of airports:
YNT to VIE: 7962; YNT to ROV: 6644; YNT to NSH: 6176; YNT to CFC: 18668; YNT to TEB: 12263; TEB to VIE: 6987; TEB to ROV: 8529; TEB to NSH: 10389; TEB to CFC: 8846; CFC to VIE: 10716; CFC to ROV: 12244; CFC to NSH: 13193; VIE to ROV: 1736; VIE to NSH: 3406; NSH to ROV: 3000.

Please calculate the shortest tour and format your answer as follows: [Airport A, Airport B, Airport C, ..., Airport A]
Identify the Minimum Vertex Cover of this network and present your answer in the following format: [UserA, UserB, UserC, UserD, ...].

---

