# OpenReview forum: "GraphArena: Evaluating and Exploring Large Language Models on Graph Computation"
_ICLR.cc/2025/Conference — ICLR 2025 Poster_

### Official Review · Reviewer_8N9w · 2024-11-02

**Soundness:** 3
**Presentation:** 3
**Contribution:** 3
**Rating:** 6
**Confidence:** 3

**Summary:**

The paper presents GraphArena, a benchmarking tool for evaluating LLMs on graph computational problems. GraphArena includes polynomial-time and NP-complete tasks and a detailed evaluation framework. The study tested over ten LLMs on 10,000 problems, finding that models often struggle with complex graphs and exhibit hallucination. It also explores some solutions: chain-of-thought prompting, instruction tuning, code writing, and scaling test-time compute.

**Strengths:**

S1: It's great to see that GraphArena includes diverse real-life datasets, addressing the limitation of benchmarks that only use synthesized graphs.

S2: The evaluation study is thorough, covering a broad spectrum of LLMs. The techniques proposed to enhance task performance are intriguing, and the results are valuable to the community.

S3: The paper is well-written and easy to read.

**Weaknesses:**

### Major weaknesses

W1: The discussion and motivation can be improved. This paper presents a useful and sound benchmark and conducts a comprehensive analysis, but the impact of these findings on practitioners and researchers is not clear. I recommend adding a discussion section in / after Section 3.2 to elaborate on the main takeaways of this paper. For example, are the findings in Section 3.2 surprising? How should we interpret the results from Section 3? What should researchers do in the future?

W2: Some experiments are not consistent. For example, SFT is only applied to Llama and Qwen, while coder is only applied to GPT and DeepSeek. Why not conducting all the proposed methods in Section 3.2 on all the models tested in Section 3.1?

W3: While I appreciate the use of real graphs in GraphArena, the connection between the tasks, findings, and these real graphs is weak. I recommend (1) including examples that demonstrate the tasks in GraphArena are relevant to real-world graph tasks and (2) providing a stronger rationale for using LLMs to solve these tasks over traditional graph algorithms.

### Minor weaknesses

M1: The many green boxes in the introduction makes this section hard to read. I recommend disabling the box highlighting or reducing citing redundant papers. For example "(Li et al., 2023c; Jin et al., 2023; Perozzi et al., 2024; Wang et al., 2023; Chen et al., 2024b)." => "(e.g., Li et al., 2023c; Jin et al., 2023; Perozzi et al., 2024)."

M2: Figure 6 comes before Figure 5.

M3: It is rather confusing to have SFT and coder in the same table, while COT and test-time compute in other different figures.

**Questions:**

Q1: It is unclear how testing LLMs on graph computational problems is a good way to assess reasoning capabilities (L48). How is reasoning capability defined?

Q2: What do "intuitive thinking" and "conscious thinking" mean for an LLM (L64)?

Q3: What are these templates? (L214) Are graphs always encoded as an edge list (figure 1)? Can one use adjacency list or adjacency matrix to represent a graph in text?

---

> ### Author Response · Authors · 2024-11-25
> **Author Response to Reviewer 8N9w (1/2)**
>
> We appreciate the reviewer's informative and constructive feedback. We provide point-to-pint responses below.
>
> >W1: ...the impact of these findings on practitioners and researchers is not clear. I recommend adding a discussion section in / after Section 3.2 to elaborate on the main takeaways of this paper. For example, are the findings in Section 3.2 surprising? How should we interpret the results from Section 3? What should researchers do in the future?
>
> The main takeaways of experimental findings are in the last two paragraphs of the introduction. We've highlighted them and added cross-references in the revision. Novel findings include:
>
> * Even leading models like GPT-4o and Claude-3.5-Sonnet face challenges with graph computation and exhibit a propensity for hallucination. This issue is more pronounced with larger graphs, more complex tasks, and smaller models.
>
> * Comparisons between LLMs and graph algorithms reveal that LLMs often resort to using greedy algorithms for addressing NP problems, yet they sometimes opt for more effective approximation algorithms.
>
> * Strategies like instruction tuning can significantly enhance performance on small-scale polynomial tasks but shows limited effectiveness on large-scale NP problems, suggesting that fine-tuning alone does not easily impart inherent advanced reasoning capabilities, such as algorithmic planning and long-context reasoning.
>
> We discussed future studies in the conclusion section:
>
> *Our findings suggest promising directions for future research, such as (1) exploring advanced finetuning techniques to enhance LLMs' code writing and algorithmic reasoning abilities, (2) developing more efficient approaches to scaling test-time compute and addressing hallucination, and (3) integrating external tools or encoder modules to better support LLMs in graph computation.*
>
> >W2: Some experiments are not consistent. For example, SFT is only applied to Llama and Qwen, while coder is only applied to GPT and DeepSeek. Why not conducting all the proposed methods in Section 3.2 on all the models tested in Section 3.1?
>
> Both code writing evaluation and instruction tuning are resource-intensive processes. Therefore, our strategy is to select representative LLMs to focus on.
>
> We chose GPT-4o and DeepSeek-V2 as our code models because they were the top two performers on OpenAI's HumanEval, a well-regarded code generation benchmark, at the time of our experiments. By focusing on their performance, we aim to capture the current upper limits of how LLMs perform in code writing for graph task solving.
>
> Fine-tuning LLMs demands significant resources, and our computational capacity only supports full-parameter fine-tuning of open-source models with fewer than 10 billion parameters. We chose Llama3-8b and Qwen2-7b because they were the leading LLMs of this scale on GSM8K, a well-known math benchmark, when we conducted our experiments. This suggests their superior math reasoning capabilities. We believe that fine-tuning these models will provide greater benefits and yield more valuable insights.
>
> >W3: While I appreciate the use of real graphs in GraphArena, the connection between the tasks, findings, and these real graphs is weak. I recommend (1) including examples that demonstrate the tasks in GraphArena are relevant to real-world graph tasks and
>
> Thank you for your suggestion. We've revised our introduction section to better motivate the the use of real graphs, the selection tasks, how they connect to reasoning, and the corresponding experimental findings.
>
> The tasks in Grapharena **have significant applications across various fields**, such as social network analysis (e.g., using common neighbor and maximum clique for community detection), bioinformatics (employing GED and MCS for molecular comparison and protein search), and operations research (utilizing shortest path and TSP for routing).
>
> **In GraphArena, we build a real-world scenario for each task** and use realistic graphs to enhance the authenticity, like flight route planning for the Traveling Salesman Problem, marking a substantial improvement over previous benchmarks that relied on synthetic graphs. **Examples for all tasks can be found in Tables 7-16 in the appendix.**
>
> >(2) providing a stronger rationale for using LLMs to solve these tasks over traditional graph algorithms.
>
> Given that not all NP-complete problems can be solved with algorithms providing exact solutions, leveraging LLMs to tackle these challenges can yield valuable insights. As shown in Figure 3, GPT-4o occasionally surpasses approximate algorithms, highlighting its potential as an alternative heuristic that complements existing approximation methods within the graph community. Additionally, using LLMs to solve these tasks has the potential to bridge the gap between classical algorithms and natural language inputs that are otherwise inaccessible. We've added these discussions in the revised related work section.

---

> > ### Author Response · Authors · 2024-11-25
> > **Author Response to Reviewer 8N9w (2/2)**
> >
> > >Q1: It is unclear how testing LLMs on graph computational problems is a good way to assess reasoning capabilities (L48). How is reasoning capability defined?
> > >Q2: What do "intuitive thinking" and "conscious thinking" mean for an LLM (L64)?
> >
> > We recognize the ambiguity of "intuitive thinking" and "conscious thinking". In our revision, we used a more precise definition of reasoning patterns: **direct algorithmic reasoning** and **meta algorithmic planning**.
> >
> > * **Direct Algorithmic Reasoning** involves *executing established algorithmic procedures*. In polynomial-time tasks, models primarily utilize this form of reasoning by following clear, deterministic steps to arrive at a solution. For instance, in the common neighbor problem, the model must (1) identify the neighbors of the first node, (2) identify the neighbors of the second node, and (3) compute their intersection. The LLM's role is to adhere strictly to the algorithmic steps while minimizing errors, thus evaluating its capability to perform direct algorithm execution and computational procedure application.
> >
> > * **Meta Algorithmic Planning** means *analyzing and selecting among different algorithms based on problem characteristics*. This type of reasoning is essential for NP-complete tasks, where there are many algorithm options. For example, the Traveling Salesman Problem with fewer than 10 nodes can be solved through exhaustive enumeration. However, as the number of nodes increases, more efficient algorithms should be considered, such as dynamic programming or approximated approaches. This requires a higher level of strategic decision-making.
> >
> >
> > >Q3: What are these templates? (L214) Are graphs always encoded as an edge list (figure 1)? Can one use adjacency list or adjacency matrix to represent a graph in text?
> >
> > **Templates for all tasks can be found in Tables 7-16 in the appendix. Graphs are encoded as an edge list for all tasks.** Furthermore, we analyzed the impact of three different graph tokenizers on GPT-4o's performance across two selected tasks:
> >
> > |GPT-4o|Shortest Distance (small)|Shortest Distance (Large)|TSP (Small)|TSP (Large)|
> > |:----|:----|:----|:----|:----|
> > |Graph Tokenizer|Fea./Hallu./Acc.|Fea./Hallu./Acc.|Fea./Hallu./Acc.|Fea./Hallu./Acc.|
> > |Edge List|**0.914/0.086/0.902**|0.788/0.192/0.720|**0.994/0.004/0.522**|0.870/0.128/0.000|
> > |Adjacency List|0.898/0.102/0.894|**0.808/0.166/0.774**|0.990/0.006/0.288|**0.918**/0.082/0.000|
> > |Adjacency Matrix|0.842/0.158/0.780|0.546/0.436/0.448|0.940/0.058/0.318|0.806/**0.012/0.016**|
> >
> > As observed, there is no universally optimal graph tokenizer. In our main experiments, we selected the edge list as our graph tokenizer because it appears to be the most stable option across various tasks and graph properties. Additionally, the edge list is the most widely used approach in prior studies, such as NLGraph and GraphWiz. In contrast, using an adjacency matrix to represent a sparse graph can introduce redundant information, which may negatively impact model performance. Due to time constraints, we are continuing our investigation in this area and will include more tasks and LLMs in the final version.
> >
> > **For Minor Weaknesses:** In the revised manuscript, we (1) reduced citing redundant papers, (2) adjusted the locations of Figures 5 and 6, and (3) splited table 2 for instruction tuning and code writing.

---

> > > ### Comment · Reviewer_8N9w · 2024-11-26
> > > **Thank you so much for the detailed response!**
> > >
> > > Thank you so much for the detailed response! It has addressed most of my concerns. I will keep my original score.

---

> > > > ### Author Response · Authors · 2024-11-28
> > > > **Author Response to Reviewer 8N9w**
> > > >
> > > > Thank you for your encouraging response and continued engagement!

---

### Official Review · Reviewer_jb83 · 2024-11-04

**Soundness:** 4
**Presentation:** 4
**Contribution:** 2
**Rating:** 8
**Confidence:** 5

**Summary:**

This paper introduces GraphArena, a benchmark designed to evaluate large language models (LLMs) on a range of classical polynomial-time and NP-complete graph algorithms. GraphArena consists of four polynomial-time tasks and six NP-complete problems set on real-world graphs sampled from various domains.
GraphArena categorizes LLM responses as correct, suboptimal, hallucinatory, or missing, and ranks overall model performance using mean reciprocal rank (MRR), top-1, and top-3 probability. The authors evaluate over ten LLMs on GraphArena and show that current SOTA models display significant limitations on the included graph reasoning tasks, especially on larger graphs. The study further examines four strategies to improve LLMs' graph reasoning abilities: chain-of-thought prompting, instruction tuning, code generation, and scaling test-time compute. Each approach yields improvements; fine-tuning models yields particularly significant improvements in accuracy, feasibility, and reducing hallucinations on both P and NP tasks for both small and large graphs.
This benchmark highlights the gap between current LLM capabilities and the demands of complex graph reasoning tasks, and contributes insight into the current understanding of LLMs' limitations and potential in graph-based reasoning.

**Strengths:**

The paper provides several NP-Complete algorithmic reasoning tasks: MCP, MIS, MVC, MCS, GED, and TSP, which are a novel inclusion in the space of benchmarking LLM performance on graph reasoning. The selected P and NP tasks offer a representative sample of graph reasoning operations. The paper does a great job of providing clear experimental results and thorough evaluation of the models on the included tasks. While the study is not the first to evaluate LLMs on real-world graph datasets, there is novelty in evaluating LLMs on complex reasoning tasks on real world graphs, especially with additional textual features.

When considering the lack of breadth available in the peer-reviewed literature of benchmarks evaluating LLM performance on graphs published before July, this work provides novel insights into the SOTA algorithmic reasoning performance.

**Weaknesses:**

The paper’s main contributions are the inclusion of real-world graphs in its dataset and textual features in both the input and desired output. However, the paper does not clearly justify evaluating LLMs on these particular tasks over those in prior benchmarks exploring LLM graph reasoning. This paper would also benefit from justifying the exclusion of neural algorithmic reasoning literature (both GNN and LLM-based), particularly the recent papers, “Benchmarking ChatGPT on Algorithmic Reasoning” (McLeish et al. 2024) and “The CLRS-Text Algorithmic Reasoning Language Benchmark” (Markeeva et al. 2024).

Claim 3: The tasks in this benchmark seem to be structured as Input-Output (IO) problems. There is limited novelty in showing that fine-tuning improves reasoning improves performance on IO tasks, as shown in “Specializing smaller language models towards multistep reasoning” (Fu et al. 2023) and other prior works. The claim of “improving graph problem-solving” should be reduced unless the authors plan to provide additional experiments with other open-source LLMs that showcase novel findings, or can provide adequate justification for this claim.

Rigorous Path-based Evaluation: The authors mention they contribute a “rigorous, path-based evaluation”, but appear to be performing exact match accuracy and then a partial accuracy metric and integrated error analysis (Missing and Hallucination). These methods are used in “Can Language Models Solve Graph Problems in Natural Language?” (Wang et al. 2023). Multiple prior methods have required models to output a listing or mapping of node IDs as a response to a query, namely “Evaluating Large Language Models on Graphs: Performance Insights and Comparative Analysis” (Liu and Wu 2023), "Can Graph Descriptive Order Affect Solving Graph Problems with LLMs?" (Ge et al. 2024), and Wang et al. 2023 on shortest path queries.

I would be willing to raise my score if the above issues are addressed.

Minor Weaknesses:
- Inclusion of synthetic graph results for the top performing LLMs would strengthen the claim that prior benchmarks fail to adequately stress test LLMs with diverse graphs
- The paper should mention the strong likelihood of the real-world graph datasets being included in the pre-training datasets of many (if not all) of the models being evaluated.
- The paper would benefit from a comparison with other recent works in LLM algorithmic reasoning, for example, Ge et al. 2024, “Transformers meet Neural Algorithmic Reasoners” (Bounsi et al. 2024), “Are Large Language Models Graph Algorithmic Reasoners?” (Taylor et al. 2024), and other recent works in the NAR field in the camera-ready version, as these papers cover a significant breadth of graph algorithmic reasoning tasks and methods for improving LLM algorithmic reasoning.
- Table 2 has several misleading bold results:
  - Lines 439-440: Qwen2-7b should be bolded for feasibility in NP small graphs
  - Lines 441-442: GPT-4o-Coder should be bolded for Polynomial large graphs
  - Lines 443-444: DeepSeek-V2 base should have bold text for accuracy in NP small graphs and feasibility in NP large graphs
- Figure 6 reference links to figure 5 for some reason
- The paper would benefit from providing a centralized list of all models evaluated and their settings, including any hyperparameter tuning

**Questions:**

- In the appendix, several prompts are shown in which the size of the graph problem presented for CoT reasoning is larger than the query problem. How did you decide on the size of graph problem to use in CoT? Is it standard across all CoT prompts? If it was tuned, please describe the process
- Is there further analysis of the types of errors that LLMs are prone to on this benchmark?

---

> ### Author Response · Authors · 2024-11-25
> **Author Response to Reviewer jb83 (1/3)**
>
> We appreciate the reviewer's informative and constructive feedback. Our responses will cover: (1) justification of the selected tasks, (2) discussion of neural algorithmic reasoning benchmarks and methods, (3) the overclaim regarding 'Improving' Graph Computation, (4) the contribution of rigorous path-based evaluation, and (5) other minor weaknesses and questions.
>
> ### 1. Justification of Selected Tasks
>
> >The paper’s main contributions are the inclusion of real-world graphs in its dataset and textual features in both the input and desired output. However, the paper does not clearly justify evaluating LLMs on these particular tasks over those in prior benchmarks exploring LLM graph reasoning.
>
> Compared with prior benchmarks, we exclude some trivial polynomial-time tasks and include more challenging NP-complete tasks. We have:
> * four polynomial-time tasks to test **direct algorithmic reasoning**, requiring models to accurately execute established algorithm procedures step-by-step.
> * six NP-complete tasks to evaluate **meta algorithmic planning**, where models must analyze problem characteristics and select suitable algorithms to excute. For example, the Traveling Salesman Problem with fewer than 10 nodes can be solved through enumeration. However, as the number of nodes increases, more efficient algorithms like dynamic programming or approximated approaches should be considered. This requires a higher level of reasoning and strategic decision-making for the model, and is overlooked by prevous benchmark study.
>
> Analogous to mathematics, polynomial-time tasks are akin to K-12 math problem, while NP-complete graph problems are more like Math Olympiad challenges. Currently, LLMs have shown the ability to solve Olympiad-level geometry and proof problems, even winning a silver medal in the recent IMO. Therefore, concentrating on these challenging tasks is crucial for understanding the limitations and potential of LLMs, providing valuable insights to guide future research in the field.
>
> **Additionally, we focus on well-known tasks with rich real-world applications**. When asked "*what are the top 10 NP-complete graph problems"*, Claude-3.5-Sonnet lists all six NP tasks we selected, considering that GED and MCS are variants of Subgraph Isomorphism. The polynomial tasks we selected are also widely recognized, with shortest path and graph diameter (multi-source shortest path) being two of the most classic algorithms in algorithm textbooks. These tasks have significant applications across various fields, including social network analysis (e.g., using common neighbor and maximum clique for community detection), bioinformatics (employing GED and MCS for molecular comparison and protein search), and operations research (utilizing shortest path and TSP for routing).
>
> * In GraphArena, we create a real-world scenario for each task and use realistic graphs to enhance the authenticity, marking a substantial improvement over previous benchmarks that relied on synthetic graphs.
>
> ### 2. Discussion of Neural Algorithmic Reasoning Benchmarks and Methods
>
> >This paper would also benefit from justifying the exclusion of neural algorithmic reasoning literature (both GNN and LLM-based), particularly the recent papers, “Benchmarking ChatGPT on Algorithmic Reasoning” (McLeish et al. 2024) and “The CLRS-Text Algorithmic Reasoning Language Benchmark” (Markeeva et al. 2024).
> >The paper would benefit from a comparison with other recent works in LLM algorithmic reasoning, for example, Ge et al. 2024, “Transformers meet Neural Algorithmic Reasoners” (Bounsi et al. 2024), “Are Large Language Models Graph Algorithmic Reasoners?” (Taylor et al. 2024), and other recent works in the NAR field...
>
> Thank you for providing these valuable references, we added the following paragraph in related work to discuss neural algorithmic reasoning.
>
> ```
> The development of neural networks capable of performing algorithmic computations has the potential to bridge the gap between classical algorithms and inputs that are otherwise inaccessible (Veličković & Blundell, 2021; Bevilacqua et al., 2023). These networks also show potential as more efficient and effective alternatives to human-crafted heuristics for combinatorial optimization tasks (Bengio et al., 2021; Georgiev et al., 2024). This field has progressed through various approaches, including graph neural networks (Khalil et al., 2017; Veličković et al., 2020), reinforcement learning (Mazyavkina et al., 2021), LLMs (Fu et al., 2023; McLeish et al., 2024; Taylor et al., 2024), and GNN-LLM hybrid models (Bounsi et al., 2024; Perozzi et al., 2024). Benchmarks like CLRS (Veličković et al., 2022) and its text-based version (Markeeva et al., 2024) assess LLMs on a wide range of general algorithmic reasoning tasks, focusing on the breadth of reasoning. In contrast, GraphArena targets challenging graph computational problems that demand more advanced reasoning skills, emphasizing the depth of reasoning.
> ```

---

> > ### Author Response · Authors · 2024-11-25
> > **Author Response to Reviewer jb83 (2/3)**
> >
> > ### 3. The overclaim regarding "Improving" Graph Computation
> >
> > > The tasks in this benchmark seem to be structured as Input-Output (IO) problems. There is limited novelty in showing that fine-tuning improves reasoning performance on IO tasks, as shown in “Specializing smaller language models towards multistep reasoning” (Fu et al. 2023) and other prior works. The claim of “improving graph problem-solving” should be reduced unless the authors plan to provide additional experiments with other open-source LLMs that showcase novel findings, or can provide adequate justification for this claim.
> >
> > We agree that the term "improving" may be overstated, so we have changed it to "exploring". Our title is now "Evaluating and **Exploring** Large Language Models on Graph Computation".
> >
> > As a benchmark study, our focus is not on proposing new architectures or methods. Instead, we aim to evaluate established strategies, such as instruction tuning, **within a broader and more diverse context that includes small/large graphs and Polynomial/NP tasks**. Our study delivers novel insights using existing technologies, such as:
> >
> > * Instruction tuning can significantly enhance performance on small-scale polynomial tasks but shows limited effectiveness on large-scale NP problems. For instance, accuracy increased from 1.4% to 7.5%, while feasibility decreased from 31.2% to 25.5% on Llama3-8b-SFT. This suggests that fine-tuning alone does not easily boost inherent advanced reasoning capabilities, such as meta algorithmic planning and long-context multi-step reasoning.
> >
> > ### 4. The Contribution of Rigorous Path-based Evaluation
> >
> > > Rigorous Path-based Evaluation: The authors mention they contribute a “rigorous, path-based evaluation”, but appear to be performing exact match accuracy and then a partial accuracy metric and integrated error analysis (Missing and Hallucination). These methods are used in “Can Language Models Solve Graph Problems in Natural Language?” (Wang et al. 2023). Multiple prior methods have required models to output a listing or mapping of node IDs as a response to a query, namely “Evaluating Large Language Models on Graphs: Performance Insights and Comparative Analysis” (Liu and Wu 2023), "Can Graph Descriptive Order Affect Solving Graph Problems with LLMs?" (Ge et al. 2024), and Wang et al. 2023 on shortest path queries.
> >
> > Thank you for bringing this to our attention. Let us clarify our evaluation process first, which differs from "performing exact match accuracy and then a partial accuracy metric".
> >
> > * Step 1: **Path Extraction**. We utilize regular expressions to extract the proposed solution path from the LLM's response (e.g., a sequence of nodes for shortest distance). If a formatted path cannot be extracted, the response is classified as missing.
> >
> > * Step 2: **Feasibility Check**. We verify if the extracted path adheres to the problem's basic requirements. For the shortest path, this involves confirming that the path connects the given starting point to the given ending point on the graph. Paths that fail this check, such as those with incorrect starting or ending points or nonexistent edges, are categorized as hallucinatory.
> >
> > * Step 3: **Optimality Verification**. We calculate the path score, such as the path length for the shortest distance, and compare it to the optimal solution obtained through exact algorithms. If the scores match, the result is optimal; otherwise, it is considered suboptimal. For tasks with multiple optimal solutions, we can check against all of them.
> >
> > While previous studies have also considered path evaluation, our proposed methods advance the field in two ways:
> >
> > * **Broader Task Coverage**. The works you mentioned (Wang et al. 2023; Liu and Wu 2023; Ge et al. 2024) **provide path evaluation only for the shortest path task** and rely on exact match answers for other tasks. In contrast, we apply our three-step path-based evaluation to all 10 tasks, which required substantial effort, including writing over 1000 lines of code for the evaluation scripts.
> >
> > * **Increased Informativeness**. Our approach not only provides detailed categorization but also enables performance ranking among different LLMs, even when all responses are suboptimal—a common scenario for NP tasks. For instance, as shown by the top 1/3 metrics for NP tasks in Table 1, identifying better suboptimal solutions can reveal the LLMs' meta algorithmic planning capabilities.

---

> > > ### Author Response · Authors · 2024-11-25
> > > **Author Response to Reviewer jb83 (3/3)**
> > >
> > > ### 5. Other Minor Weakness and Questions
> > >
> > > >Inclusion of synthetic graph results for the top performing LLMs would strengthen the claim that prior benchmarks fail to adequately stress test LLMs with diverse graphs
> > >
> > > When evaluating GPT-4o on two polynomial-time tasks, we observe that it achieves significantly higher accuracy and reduced hallucination rates on synthetic Erdős-Rényi graphs compared to real-world graphs of similar scale. This performance difference suggests that synthetic graphs may not adequately capture the complex topological features and irregular structures inherent in real-world networks. Due to time constraints, we are continuing our investigation in this area and will include more tasks and LLMs in the final version.
> > >
> > > |TasksGPT-4o|Neighbor (Small)|Neighbor (Large)|Distance (small)|Distance (large)|
> > > |:----|:----|:----|:----|:----|
> > > |Graph Type|Fea./Hallu./Acc.|Fea./Hallu./Acc.|Fea./Hallu./Acc.|Fea./Hallu./Acc.|
> > > |Real-world Graphs|0.838/0.088/0.778|0.740/0.140/0.626|0.922/0.078/0.904|0.760/0.224/0.680|
> > > |Erdős-Rényi Graphs|**0.996/0.004/0.986**|**0.984/0.016/0.730**|**0.980/0.02/0.946**|**0.864/0.146/0.702**|
> > >
> > >
> > > >The paper should mention the strong likelihood of the real-world graph datasets being included in the pre-training datasets of many (if not all) of the models being evaluated.
> > >
> > > Although the real-world graph datasets used in our evaluation might have been part of the pre-training data for the evaluated LLMs, this overlap does not undermine the validity of our assessment. This is because GraphArena generates problems by randomly sampling subgraphs from networks with millions of nodes, making it highly unlikely for identical problem instances to appear during pre-training. For instance, even with the relatively small OpenFlights graph, which consists of 3,390 nodes, selecting just 5 nodes for a TSP problem results in over $10^15$ possible combinations. Consequently, while LLMs may have encountered the underlying graphs during pre-training, the specific problem instances in our evaluation are essentially novel.
> > >
> > >
> > > >The paper would benefit from providing a centralized list of all models evaluated and their settings, including any hyperparameter tuning.
> > >
> > > We have detailed all model settings in the experimental setup section. For all main experiments, we use a temperature of 0.1 and keeping all other parameters at their default values. No hyperparameter tuning was performed, as our goal was to evaluate the models' base capabilities under standard conditions. We have also added the default fine-tuning hyperparameters used in the revision.
> > >
> > >
> > > >In the appendix, several prompts are shown in which the size of the graph problem presented for CoT reasoning is larger than the query problem. How did you decide on the size of graph problem to use in CoT? Is it standard across all CoT prompts? If it was tuned, please describe the process.
> > > We did not tune the CoT examples.
> > >
> > >
> > > For the $k$-shot CoT prompting experiments presented in Figure 6, we randomly selected 4 examples, each containing 4 to 5 nodes. In comparison, the scale of query problems ranges from 4 to 30 nodes. We manually crafted detailed step-by-step solutions for these examples, as shown in Tables 7 and 8. Finally, we randomly selected $k$ examples from this pool to demonstrate before presenting the query problem.
> > >
> > >
> > > For the main experiments, to help the LLM better understand the solution format, we randomly generated 100 question-answer path pairs (see Tables 9-16) as examples. For the $i$th query problem, we demonstrated the $i$%100th example. These examples follow the same scale distribution as our small-scale dataset and may sometimes be larger than the query problem.
> > >
> > > >Is there further analysis of the types of errors that LLMs are prone to on this benchmark?
> > >
> > > In our paper, we identify three categories of errors: missing, hallucination, and suboptimal. Among these, LLMs are particularly prone to hallucination errors, especially as graph size increases, as illustrated in Figure 4. Hallucination errors can be further subdivided into various types, such as misinterpretation of graph structure or misunderstanding of problem requirements. However, localizing these specific errors is challenging and varies with each problem, so we have left this for future work.

---

> ### Comment · Reviewer_jb83 · 2024-11-26
>
> My concerns have been addressed thoroughly and I have updated my score.
>
>
> While including the references I mentioned in my review is an improvement, the above paragraph does not quite address my original concern. The field of neural algorithmic reasoning is quite rich, and multiple benchmarks are already established (Veličković et al., 2022; McLeish et al., 2024; Taylor et al., 2024; Markeeva et al., 2024). I was mainly asking for a paragraph describing what your paper provides that is missing from these benchmarks. Please give a view of how your paper fills the gaps in the current NAR field (hint: neither McLeish et al., 2024 nor Taylor et al., 2024 involve reinforcement learning).
>
> I would recommend that the above evaluation details be included in the camera-ready version of the work (preferably in the main text). It is rare for benchmarks in this space to evaluate answers as graphs, and this could serve as a model for future work as well.

---

> > ### Author Response · Authors · 2024-11-28
> > **Author Response to Reviewer jb83**
> >
> > Thank you for your active engagement, encouraging response, and valuable suggestions!
> >
> > > I would recommend that the above evaluation details be included in the camera-ready version of the work (preferably in the main text). It is rare for benchmarks in this space to evaluate answers as graphs, and this could serve as a model for future work as well.
> >
> > We have incorporated the evaluation processes and details in section 2.3 of the revised manuscript.
> >
> >
> > > I was mainly asking for a paragraph describing what your paper provides that is missing from these benchmarks. Please give a view of how your paper fills the gaps in the current NAR field (hint: neither McLeish et al., 2024 nor Taylor et al., 2024 involve reinforcement learning).
> >
> >
> > We have revised the third paragraph in our introduction section to highlight the limitations of both existing graph and NAR benchmarks. These limitations align with the three key contributions of GraphArena, which are detailed in the following paragraph: realistic graph collection, curated task selection, and rigorous path-based evaluation.
> >
> > Here is the revise paragraph:
> >
> > ---
> >
> > ...While several graph problem-solving benchmarks have been established, such as NLGraph (Wang et al., 2023) and GraphQA (Fatemi et al., 2023) , **as well as algorithic reasoning benchmarks like CLRS-text (Markeeva et al., 2024) and MAGMA (Taylor et al., 2024)**, we identify critical limitations in the existing landscape. First, these benchmarks are **predominantly built on synthetic graphs**, such as those generated by the Erdos-Rényi (ERDdS & R&wi, 1959) models, which may not adequately reflect real-world diversity. Second, the tasks within these benchmarks are generally confined to basic structural understanding like connectivity checks and **algorithm execution like breadth-first search**. They largely overlook the assessment of **higher-order reasoning skills, such as problem abstration and simplification, strategy comparision and selection, etc. These capabilities are better demonstrated through solving more complex and open-ended tasks, such as NP-complete problems**. Third, **current evaluation methods typically rely on string matching between the final answers and the responses**, allowing models to succeed through guesswork rather than demonstrating a genuine understanding of the underlying logic. Such methods also lack a nuanced categorization of failure modes, thereby limiting the depth of insights derived from the evaluation metrics.
> >
> > ---
> >
> > We are still learning from the neural algorithmic reasoning field, particularly the CLRS benchmark series. With their extensive algorithm coverage and detailed reasoning traces, these benchmarks can serve as valuable resources for continual pretraining of LLMs to enhance relevant reasoning capabilities. In contrast, the GraphArena corpus is more suited for the fine-tuning stage due to its more challenging and domain-specific problems. Regarding reinforcement learning, this aspect has not been addressed in the current paper. Should we gain further insights, we will include them in the camera-ready version.

---

### Official Review · Reviewer_H9rP · 2024-11-04

**Soundness:** 4
**Presentation:** 4
**Contribution:** 3
**Rating:** 8
**Confidence:** 3

**Summary:**

- The paper introduces GraphArena, a LLM benchmark for measuring graph reasoning.
    - GraphArena is based on real world graphs, including knowledge graphs, social networks, and molecular structures
    - The tasks constructed on these graphs capture a spectrum of different reasoning skills
    - The authors introduce a path-specific evaluation methodology which allows for finegrained measurement of LLM responses
- In addition to describing GraphArea, the paper compares the performance of different LLMs to a broad range of graph-based methods.
- Finally, the authors explore a range of strategies to improve performance on GraphArea (finetuning, prompting, scaling test time compute).

**Strengths:**

- I think the benchmark is extremely well constructed.
    - It is based on realistic data
    - It has a spectrum of difficulty, so different researchers may focus on different subsets of samples/tasks based on the types of models they are studying
    - Similarly, the evaluation protocol applied make sense, and section 2.3 does a good job of motivating it.
- The experiments are rigorous and comprehensive. I think the exploration of methods for improving performance is well-done and rounds out the paper nicely.

**Weaknesses:**

I don't think the paper has any significant weaknesses. My only quibble is that while the graphs are drawn from the real-world, the questions themselves are not. But I think this is pretty minor, and shouldn't affect what ultimately seems like an interesting paper with a meaningful contribution. If the goal is to measure graph reasoning skills, then I think relying on artificial problems is perfectly acceptable.

**Questions:**

Are there any performance trends across different graph domains (e.g., DBPedia vs PubChemQC)? I wonder if differences in how graphs are tokenized by the model influence performance.

---

> ### Author Response · Authors · 2024-11-25
> **Author Response to Reviewer H9rP**
>
> We sincerely thank the reviewer for your supportive and constructive feedback.
>
> >I don't think the paper has any significant weaknesses. My only quibble is that while the graphs are drawn from the real-world, the questions themselves are not. But I think this is pretty minor, and shouldn't affect what ultimately seems like an interesting paper with a meaningful contribution. If the goal is to measure graph reasoning skills, then I think relying on artificial problems is perfectly acceptable.
>
> We recognize the limitations of template-based question generation. However, when human experts or LLMs create and rewrite questions, they can inadvertently alter the underlying graph structure, leading to mismatched answers. These types of errors in question generation are difficult to detect.
>
>
> Given our primary objective of evaluating LLMs' graph reasoning abilities, we chose to maintain consistent question formats in this study. We plan to explore greater question diversity in future research. We added a discussion of this in the revised conclusion.
>
>
> >Are there any performance trends across different graph domains (e.g., DBPedia vs PubChemQC)? I wonder if differences in how graphs are tokenized by the model influence performance.
>
> It is an interesting problem. We explored three graph domains for the shortest path task: knowledge graphs (DBPedia), molecular graphs (PubChemQC), and synthetic graphs (Erdős–Rényi model). Here are the results:
>
> |GPT-4o|Shortest Distance (small)|Shortest Distance (large)|
> |:----|:----|:----|
> |Graph Type|Fea./Hallu./Acc.|Fea./Hallu./Acc.|
> |Knowledge graphs |0.922/0.078/0.904|0.760/0.224/0.680|
> |Molecular Graphs|**0.992/0.008/0.956**|0.858/0.122/0.690|
> |Synthetic Graphs|0.980/0.020/0.946|**0.864/0.146/0.702**|
>
> Based on GPT-4o's performance, knowledge graphs seem to be more challenging than molecular and synthetic graphs. This suggests knowledge graphs may have more complex topological features. Indeed, calculating the shortest distance in knowledge graphs is a more well-defined task, as it involves quantifying the minimum number of hops between two knowledge entities.
>
>
> Additionally, we examined the impact of three different graph tokenizers on GPT-4o's performance:
>
> |GPT-4o|Shortest Distance (small)|Shortest Distance (Large)|TSP (Small)|TSP (Large)|
> |:----|:----|:----|:----|:----|
> |Graph Tokenizer|Fea./Hallu./Acc.|Fea./Hallu./Acc.|Fea./Hallu./Acc.|Fea./Hallu./Acc.|
> |Edge List|**0.914/0.086/0.902**|0.788/0.192/0.720|**0.994/0.004/0.522**|0.870/0.128/0.000|
> |Adjacency List|0.898/0.102/0.894|**0.808/0.166/0.774**|0.990/0.006/0.288|**0.918**/0.082/0.000|
> |Adjacency Matrix|0.842/0.158/0.780|0.546/0.436/0.448|0.940/0.058/0.318|0.806/**0.012/0.016**|
>
> As observed, there is no universally optimal graph tokenizer. In our main experiments, we selected the edge list as our graph tokenizer because it appears to be the most stable option across various tasks and graph properties. Additionally, the edge list is the most widely used approach in prior studies, such as NLGraph and GraphWiz. In contrast, using an adjacency matrix to represent a sparse graph can introduce redundant information, which may negatively impact model performance.

---

### Official Review · Reviewer_Mrvi · 2024-11-05

**Soundness:** 3
**Presentation:** 3
**Contribution:** 3
**Rating:** 5
**Confidence:** 4

**Summary:**

The paper introduces GraphArena, a benchmarking tool designed to evaluate LLMs on graph computation tasks.
It includes ten LLMs with polynomial-time and NP-complete graph problems. The authors claim to go beyond existing benchmarks by including real-world graph structures, introducing path-based evaluation criteria, and testing various improving-performance methods.

**Strengths:**

1. It includes a diverse set of graph tasks, ranging from simple to complex. It provides a useful evaluation tool for understanding the reasoning capabilities of LLMs on graph problems.
2. It also includes real-world graphs, which can help evaluate the models' performance on practical applications.
3. The paper compares LLMs with traditional graph algorithms, providing insights into the strengths and weaknesses of LLMs in graph computation tasks.

**Weaknesses:**

1. The analysis is surface-level and lacks in-depth insights into the models' reasoning capabilities. I also have questions about the selected tasks, such as why these specific tasks are chosen and how they represent real-world applications.
2. While the authors position GraphArena as an improvement over previous benchmarks, the degree of novelty in terms of task design and dataset is somewhat unclear. Could the authors provide a more detailed comparison with existing benchmarks?
3. The definition of small/large graphs are different across tasks, which might introduce inconsistencies in the evaluation. It would be helpful to standardize the graph sizes for a fair comparison across tasks.
4. For the evaluation metrics (Feasibility, Hallucination...), there is no clear explanation of how these metrics are calculated.
5. The authors introduce System 1 and System 2 thinking to describe task complexity, but the connection is weak and lacks follow-through. This framing feels more like an afterthought without a clear motivation and in-depth analysis in the later sections.

**Questions:**

1. Could you elaborate on how the categories (correct, suboptimal, hallucinatory, missing) are determined in a more complex, multi-step task like the Traveling Salesman Problem? How does the evaluation framework ensure consistent scoring across models?
2. How are the polynomial-time and NP-complete tasks chosen, and do they represent a balanced challenge across diverse graph properties (e.g., density, connectivity)?
3. When using code generation to solve graph problems, how is the correctness of the generated code verified?

---

> ### Author Response · Authors · 2024-11-25
> **Author Response to Reviewer Mrvi (1/3)**
>
> We appreciate the reviewer’s detailed and constructive feedback. Our responses will cover: (1) the evaluation process, (2) the criteria for task selection, (3) the analysis of reasoning capabilies, (4) the rationale for setting different graph scales, and (5) the novelty compared with existing benchmarks.
>
> ### 1. The Evaluation Process
>
> > Question #1: Could you elaborate on how the categories (correct, suboptimal, hallucinatory, missing) are determined in a more complex, multi-step task like the Traveling Salesman Problem? How does the evaluation framework ensure consistent scoring across models?
> > Weakness #4: For the evaluation metrics (Feasibility, Hallucination...), there is no clear explanation of how these metrics are calculated.
>
> Our evaluation consists of three systematic steps: path extraction, feasibility check, and optimality check. To illustrate, consider the following TSP problem:
>
> ```plain
> You are required to solve the Travelling Salesman Problem for an undirected flight route network. Your objective is to determine the shortest possible route that visits each of the listed airports exactly once and returns to the starting point.
> **Problem to Solve**
> Airports to visit: VPY, AAE, BGA, YWB.
> Travel distances (in kilometers) between each pair of airports:
> - VPY to YWB: 16285
> - VPY to AAE: 9488
> - VPY to BGA: 13255
> - AAE to YWB: 7807
> - AAE to BGA: 9332
> - BGA to YWB: 6575
> Please calculate the shortest tour and format your answer as follows: [Airport A, Airport B, Airport C, ..., Airport A]
> ```
>
>
> We conduct the following three-step evaluation:
>
> * Step 1: **Path Extraction**. We use regular expressions to **extract the proposed solution path from the LLM's response (a sequence of airports for TSP)**. If a valid path cannot be extracted (due to formatting issues, no response, etc.), the response is categorized as **missing**. In over 99% of cases, the route can be extracted.
>
> * Step 2: **Feasibility Check**. We write scripts to **verify if the extracted path adheres to the problem's basic requirements**. For TSP, this means visiting each node exactly once and returning to the starting point. Failed routes are categorized as **hallucinatory**, such as:
>
>    * [VPY, BGA, VPY, YWB, AAE, VPY] (repeated visits)
>
>    * [VPY, BGA, YWB, AAE] (no return)
>
>    * [VPY, BGA, YWB, VPY] (missing airport AAE)
>
> * Step 3: **Optimality Verification**. For feasible paths, we **compute the path score (e.g., total route length for TSP)**. This score is compared to the optimal solution obtained by exhausted search. If they match, the result is **optimal**; otherwise, it is **suboptimal**. Examples are:
>
>    * [VPY, BGA, AAE, YWB, VPY] total distance = 46779 km, suboptimal
>
>    * [VPY, AAE, YWB, BGA, VPY] total distance = 37125 km, optimal
>
>
> **Our evaluation ensures consistent scoring as the same process is applied to all LLMs**. Each task has unique feasibility and optimality requirements, as outlined in Section 2.2 and Appendix A. We provided a more nuanced assessment compared to previous studies focusing mainly on correctness,. Even when all responses are suboptimal (common in NP problems), we can still rank LLMs based on their path score (e.g., route length in TSP).
>
>
> > Question #3: When using code generation to solve graph problems, how is the correctness of the generated code verified?
>
> We prompt LLMs to write code that outputs the solution path. The generated code is executed to print the path. If the code fails, the response is marked as **missing**. Otherwise, we evaluate the printed path using the same process mentioned above.

---

> ### Author Response · Authors · 2024-11-25
> **Author Response to Reviewer Mrvi (2/3)**
>
> ### 2. Task Selection Criteria
>
> >Question #2: How are the polynomial-time and NP-complete tasks chosen...
> >Weakness #1: Why these specific tasks are chosen and how they represent real-world applications.
>
> While there is no gold standard for task selection, we focus on two main criteria:
>
> **(1) well-known tasks with rich real-world applications**. When asked "*what are the top 10 NP-complete graph problems"*, **Claude-3.5-Sonnet lists all six NP tasks we selected**, noting that GED and MCS are variants of Subgraph Isomorphism. The polynomial tasks we selected are also widely recognized, with shortest path and graph diameter (multi-source shortest path) being two of the most classic algorithms found in algorithm textbooks.
>
> * These tasks **have significant applications across various fields**, including social network analysis (e.g., using common neighbor and maximum clique for community detection), bioinformatics (employing GED and MCS for molecular comparison and protein search), and operations research (utilizing shortest path and TSP for routing).
>
> * In GraphArena, **we build a real-world scenario for each task** and use realistic graphs to enhance the authenticity, like flight route planning for the TSP example mentioned earlier, marking a substantial improvement over previous benchmarks that relied on synthetic graphs.
>
>
> **(2) challenging tasks that assess multi-dimensional reasoning skills.** Compared with previous studies, we exclude trivial polynomial-time tasks like node counting and edge existence that can be directly solved in one step, and instead include more NP tasks. Unlike polynomial tasks with fixed solutions, NP tasks generally offer multiple solution paths. The model must determine an appropriate algorithm (e.g., exhaustive enumeration or approximation) based on the problem's structure and scale, which requires more advanced reasoning capabilities and problem-solving skills.
>
>
> Since not all NP-complete problems can be solved using algorithms that provide exact solutions, leveraging LLMs to address these challenges can offer valuable insights. As illustrated in Figure 3, GPT-4o occasionally outperforms approximate algorithms, demonstrating its potential to serve as an alternative heuristic that complements existing approximation methods in the graph community. Moreover, the search space for NP problems is generally much larger than that for polynomial-time problems. As a result, evaluating LLMs on these tasks provides a more thorough and comprehensive assessment of their reasoning capabilities.
>
> >Do they represent a balanced challenge across diverse graph properties (e.g., density, connectivity)?
>
> The inclusion of five different graph sources (DBLP, Social Network, DBpedia, Openflights, and PubChemQC) greatly improve the diversity compared with previous benchmarks that work on systhetic graphs. We added the statistics of Density and Edge connecteivity as you suggested in the task statistics table 4. As we can see, different tasks has very different Density (range from 0.23 to 1) and edge connectivity (range from 0.26 to 8.46).
>
> ||Neighbor|Distance|Component|Diameter|MCP|MIS|MVC|MCS|GED|TSP|
> |:----|:----|:----|:----|:----|:----|:----|:----|:----|:----|:----|
> |Density|0.44|0.28|0.23|0.31|0.52|0.30|0.31|0.30|0.29|1.00|
> |Edge Connectivity|1.37|0.61|0.26|1.16|1.70|1.08|1.17|1.04|1.03|8.46|
>
> The following table compared GPT-4o's performance on our real-world graphs versus synthetic Erdős-Rényi graphs of similar scale. It also highlights the diversity and complexity of authentic graph structures we used.
>
> |GPT-4o|Neighbor (Small)|Neighbor (Large)|Distance (small)|Distance (large)|
> |:----|:----|:----|:----|:----|
> |Graph Type|Fea./Hallu./Acc.|Fea./Hallu./Acc.|Fea./Hallu./Acc.|Fea./Hallu./Acc.|
> |Real-world Graphs|0.838/0.088/0.778|0.740/0.140/0.626|0.922/0.078/0.904|0.760/0.224/0.680|
> |Erdős-Rényi Graphs|**0.996/0.004/0.986**|**0.984/0.016/0.730**|**0.980/0.02/0.946**|**0.864/0.146/0.702**|

---

> > ### Author Response · Authors · 2024-11-25
> > **Author Response to Reviewer Mrvi (3/3)**
> >
> > ### 3. The Connection with Reasoning
> >
> > >Weakness #1: The analysis is surface-level and lacks in-depth insights into the models' reasoning capabilities...
> > >Weakness #5: The authors introduce System 1 and System 2 thinking to describe task complexity, but the connection is weak and lacks follow-through...
> >
> > We recognize the ambiguity of "system 1 and system 2 thinking". In our revision, we used a more precise characterization of reasoning patterns: **direct algorithmic reasoning** and **meta algorithmic planning**.
> >
> > * **Direct Algorithmic Reasoning** involves *executing established algorithmic procedures*. In polynomial-time tasks, models primarily utilize this form of reasoning by following clear, deterministic steps to arrive at a solution. For instance, in the common neighbor problem, the model must (1) identify the neighbors of the first node, (2) identify the neighbors of the second node, and (3) compute their intersection. The LLM's role is to adhere strictly to the algorithmic steps while minimizing errors, thus evaluating its capability to perform direct algorithm execution and computational procedure application.
> >
> > * **Meta Algorithmic Planning** means *analyzing and selecting among different algorithms based on problem characteristics*. This type of reasoning is essential for NP-complete tasks, where there are many algorithm options. For example, the Traveling Salesman Problem with fewer than 10 nodes can be solved through exhaustive enumeration. However, as the number of nodes increases, more efficient algorithms should be considered, such as dynamic programming or approximated approaches. This requires a higher level of strategic decision-making.
> >
> >
> > These refined definitions enhance the connection with subsequent empirical analysis, which reveals the following key findings:
> >
> > * LLMs perform significantly better on polynomial-time tasks than on NP-complete tasks, indicating that they excel more in direct algorithmic reasoning than in meta planning.
> >
> > * LLMs often struggle with hallucinations during algorithm execution, particularly on large-scale graphs and more complex tasks where the reasoning steps are extensive.
> >
> > * The comparison with classic graph algorithms reveals that LLMs often resort to using greedy algorithms for addressing NP problems, yet they sometimes opt for more effective approximation algorithms.
> >
> > * Strategies such as instruction tuning, code writing, and scaling test-time computation enhance direct algorithmic reasoning capabilities more effectively than meta algorithmic planning capabilities.
> >
> >
> > ### 4. Different Graph Scales
> >
> > >Weakness #3: The definition of small/large graphs are different across tasks, which might introduce inconsistencies...
> >
> > Our task-specific size calibration follows established practices in previous studies (e.g., NLGraph, GraphWiz). While standardizing graph sizes seems reasonable, it would actually impair difficulty calibration. Different task complexities inherently demand different scale considerations:
> >
> > * For NP-complete tasks (e.g., TSP), even graphs with 20 nodes can be computationally intractable, as finding the exact answer becomes prohibitively time-consuming at larger scales.
> >
> > * For polynomial-time tasks (e.g., common neighbors), graphs with 50 nodes remain computationally manageable.
> >
> > As evidenced in Table 1, LLMs already show significantly lower performance on NP-complete tasks despite their smaller graph sizes. Enforcing uniform graph sizes would create a larger performance gap and fail to meaningfully interpret model capabilities.
> >
> > In the revision, we highlighted that our use of ``small/large'' graphs are relative and task-specific, determined by each problem's computational complexity rather than absolute node counts.
> >
> > ### Novelty
> >
> > >Weakness #2: While the authors position GraphArena as an improvement over previous benchmarks, the degree of novelty in terms of task design and dataset is somewhat unclear...
> >
> > All our contributions are detailed in paragraphs 4-7 of our introduction, as well as in prior responses. To summarize, in terms of benchmark construction, our contributions include: (1) the assembly of a realistic graph collection, (2) a comprehensive selection of Polynomial-time and NP-complete tasks, and (3) a rigorous path-based evaluation framework.
> >
> > For empirical evaluation, our contributions are: (1) a comparative analysis between LLMs and various model categories, including GNNs and traditional graph algorithms, and (2) a novel investigation into the hallucination phenomenon, and (3) the exploration of diverse strategies to enhance LLMs' graph reasoning capabilities.

---

> > > ### Comment · Reviewer_Mrvi · 2024-11-27
> > >
> > > Thank you to the authors for their detailed responses. While I appreciate the effort to address my concerns, I still have some concerns. The evaluation of generated code is solely based on success/fail metrics, which may overlook various intermediate issues. For example:
> > > - Logical flaws
> > > - Efficiency concerns
> > > - Readability and maintainability
> > > - Edge case handling
> > >
> > > However, the authors' contributions bring value to the field. I am raising my score to 5.

---

> > > > ### Author Response · Authors · 2024-11-28
> > > > **Author Response to Reviewer Mrvi**
> > > >
> > > > Thank you for your continued engagement and thoughtful consideration of our work!
> > > >
> > > > We acknowledge that the evaluation of code generation in our benchmark is quite preliminary and is intended merely to demonstrate the potential of this direction. The current evaluation focuses on how many test cases an LLM can pass out of 1,000 examples, with a one-minute limit per example, which partially considers corner cases and efficiency. However, more advanced aspects, such as logical flaws, readability, and maintainability, are overlooked.
> > > >
> > > > While code generation is not the main focus of GraphArena's development, it remains a promising area for future exploration. For instance, investigating whether LLMs with specialized training can identify improved search methods or heuristics is a valuable pursuit. GraphArena offers a platform and foundation to support such studies, though human involvement may be necessary for comprehensive evaluation.
> > > >
> > > > The following case studies reveal that the code generated by LLMs primarily relies on brute-force enumeration and greedy-based heuristics, often overlooking other approaches. Consider the maximum independent set problem in GraphArena:
> > > >
> > > > - Case 1. When the graph size is small, GPT-4o-coder employs a brute-force approach.
> > > >
> > > > ```
> > > > import itertools
> > > >
> > > > # Users and friendships
> > > > users = ["Gwendolyn Russell", "Ashley Boyd", "Thomas Perkins", "Matthew Bennett II", "Ian Soto"]
> > > > friendships = [("Gwendolyn Russell", "Thomas Perkins"),
> > > >                ("Ashley Boyd", "Thomas Perkins"),
> > > >                ("Thomas Perkins", "Ian Soto"),
> > > >                ("Thomas Perkins", "Matthew Bennett II")]
> > > >
> > > > # Function to check if a subset is independent
> > > > def is_independent(subset, friendships):
> > > >     for u1, u2 in itertools.combinations(subset, 2):
> > > >         if (u1, u2) in friendships or (u2, u1) in friendships:
> > > >             return False
> > > >     return True
> > > >
> > > > # Find the maximum independent set
> > > > max_independent_set = []
> > > > for r in range(1, len(users) + 1):
> > > >     for subset in itertools.combinations(users, r):
> > > >         if is_independent(subset, friendships):
> > > >             if len(subset) > len(max_independent_set):
> > > >                 max_independent_set = subset
> > > >
> > > > # Print the result
> > > > print(list(max_independent_set))
> > > > ```
> > > >
> > > >
> > > > - Case 2. In a graph with 20 nodes, GPT-4o-coder adopted a greedy approach utilizing the maximum degree heuristic.
> > > >
> > > > ```python
> > > > def maximum_independent_set(users, friendships):
> > > >     # Create a graph from the users and friendships
> > > >     graph = {user: set() for user in users}
> > > >     for u, v in friendships:
> > > >         graph[u].add(v)
> > > >         graph[v].add(u)
> > > >
> > > >     # Sort nodes by degree (number of connections)
> > > >     sorted_users = sorted(graph, key=lambda x: len(graph[x]))
> > > >
> > > >     # Initialize the independent set
> > > >     independent_set = set()
> > > >
> > > >     # While there are nodes left in the graph
> > > >     while sorted_users:
> > > >         # Select the node with the smallest degree
> > > >         node = sorted_users.pop(0)
> > > >
> > > >         # Add it to the independent set
> > > >         independent_set.add(node)
> > > >
> > > >         # Remove the node and its neighbors from the graph
> > > >         neighbors = graph[node]
> > > >         for neighbor in neighbors:
> > > >             if neighbor in sorted_users:
> > > >                 sorted_users.remove(neighbor)
> > > >
> > > >         # Remove the node from the graph
> > > >         del graph[node]
> > > >
> > > >     return list(independent_set)
> > > >
> > > > # Define the users and friendships
> > > > users = [
> > > >     "Timothy Hall", "Brianna Morton", "David Walker", "Debra Cook", "Danielle Hoffman",
> > > >     "Patrick Ellis", "Jim Cruz", "Bryan Mcintosh", "Kevin Smith", "Stacy Lindsey",
> > > >     "Denise Gregory", "Jessica Hahn", "Michael Rogers", "Miguel Wilson", "Elizabeth Washington",
> > > >     "Pamela Rodriguez", "Caitlin Harris", "Craig Hall", "Denise Ponce", "Casey Martin"
> > > > ]
> > > >
> > > > friendships = [
> > > >     ("Timothy Hall", "Danielle Hoffman"), ("Timothy Hall", "Brianna Morton"), ("Timothy Hall", "Craig Hall"),
> > > >     ("Timothy Hall", "Michael Rogers"), ("Brianna Morton", "David Walker"), ("Brianna Morton", "Casey Martin"),
> > > >     ("David Walker", "Patrick Ellis"), ("Debra Cook", "Patrick Ellis"), ("Patrick Ellis", "Casey Martin"),
> > > >     ("Patrick Ellis", "Denise Ponce"), ("Patrick Ellis", "Jim Cruz"), ("Patrick Ellis", "Bryan Mcintosh"),
> > > >     ("Patrick Ellis", "Kevin Smith"), ("Patrick Ellis", "Stacy Lindsey"), ("Patrick Ellis", "Denise Gregory"),
> > > >     ("Patrick Ellis", "Jessica Hahn"), ("Patrick Ellis", "Michael Rogers"), ("Patrick Ellis", "Miguel Wilson"),
> > > >     ("Patrick Ellis", "Elizabeth Washington"), ("Patrick Ellis", "Pamela Rodriguez"), ("Patrick Ellis", "Caitlin Harris"),
> > > >     ("Jim Cruz", "Denise Gregory"), ("Jim Cruz", "Pamela Rodriguez")
> > > > ]
> > > >
> > > > # Find the maximum independent set
> > > > result = maximum_independent_set(users, friendships)
> > > > print(result)
> > > > ```

---

### Author Response · Authors · 2024-11-25
**Revision Summary**

Dear Reviewers,

We sincerely appreciate your constructive feedback and recognition of our contributions. We have carefully reviewed each comment and provided point-to-pint responses. Based on your suggestions, we have made the following revisions to our manuscript:

### Title

* Changed the word 'improving' to 'exploring' to avoid over-claiming. Now the title is "Evaluating and Exploring Large Language Models on Graph Computation".


### Introduction

* Revised the description of LLMs' reasoning capabilities. We replaced ambiguous terms like 'system 1/intuitive thinking' and 'system 2/conscious thinking' with more precise terms such as 'direct algorithmic reasoning' and 'meta algorithmic planning.' The subsequent content of the paper has been adjusted accordingly.

* Added justification for evaluating LLMs on these specific tasks, highlighting the differences from prior benchmarks exploring LLM graph reasoning.

* Revised the presentation of experimental findings to ensure consistency in analysis.

* Reduced citing redundant papers.


### Benchmark Construction

* Provided more motivations for selecting these specific tasks for evaluation.

* Revised the three-step evaluation process for clarity and increased rigor.

* Included the rationale for using task-specific graph scales in problem generation.

* Added information on density and edge connectivity in the task statistics presented in Table 4.


### Experiments

* Made minor corrections to Figures 5 and 6, as well as Tables 2 and 3.

* Included hyperparameter configurations for instruction tuning.


### Related Work

* Added a paragraph discussing the background and advancements in neural algorithmic reasoning.

---

### Meta-Review · Area_Chair_FnCk · 2024-12-17

**Metareview:**

This paper proposes GraphArena, which is a benchmarking to evaluate LLMs in their ability to measure graph reasoning.  The benchmark is based on knwoledge, social network and molecular graphs, and the graph tasks are diverse.  Moreover, the authors introduce a scoring setup which tiers LLM outputs between correct, suboptimal, hallucinatory or missing.  The work complements other past benchmarks on graph reasoning tasks.

Reviewers leaned positively towards this work and felt this resource would be a value-add to the community, hence the accept recommendation.

There are a few concerns which were raised that the authors should take care to fold into the revision:

- Further rationale around the choice of tasks and their alignment to real-world applications (Mrvi, H9rP, 8N9w)

- Positioning with respect to neural algorithmic reasoning and some of the more recent related works from 2024 (jb83)

**Additional Comments On Reviewer Discussion:**

Authors addressed several concerns in the rebuttal, overlapping with those above.  A paragraph on neural algorithmic reasoning was added.  Moreover, authors improved benchmark construction details as well as experimental details in line with reviewers' questions.  Moreover, authors amended the title to avoid overclaims and revised the introduction to promote clarity and consistency around the use of terminology like algorithmic reasoning and planning.

---

### Decision · Program_Chairs · 2025-01-22

Accept (Poster)